# Context-Constrained Transfer Learning for Tabular Foundation Models via Data Distillation

**Abstract**

Tabular Foundation Models (TFMs) have demonstrated strong empirical performance as black-box inference engines through in-context learning. However, their use in transfer learning is limited by two obstacles: strict context-size constraints and sensitivity to distribution shifts between source and target tasks. Directly pooling heterogeneous source data can therefore lead to negative transfer. To address these challenges, we propose Context-Constrained Transfer Learning via ANchoring and DIstillation (TL-ANDI), a posterior-aware distillation framework for TFMs. TL-ANDI constructs a compact source context by solving a budget-constrained optimal transport problem whose cost jointly measures target covariate coverage and posterior compatibility. The selected anchor samples are then equipped with locally distilled labels and combined with a residual calibration step using target data. We establish two formal guarantees: an oracle inequality showing that the distilled source context is target-compatible over the observed test region whenever transferable source regions exist, and a validation-based no-negative-transfer guarantee for the complete black-box procedure. Extensive empirical evaluations on simulated and real-world datasets demonstrate that TL-ANDI robustly mitigates negative transfer and improves over naive transfer learning baselines in both regression and classification problems.

**Keywords:** transfer learning, tabular foundation model, optimal transport, data distillation, posterior shift

**Mathematics Subject Classification (2020):** 62G05, 68T05

## 1 Introduction

Tabular Foundation Models (TFMs), such as TabPFN (Hollmann et al., 2025) and Limix (Zhang et al., 2025) have recently achieved remarkable success in tabular data analysis, largely driven by the paradigm of In-Context Learning (ICL). By taking inputs that include the training data $(X^{(\mathrm{tr})}, \boldsymbol{y}^{(\mathrm{tr})}) \in \mathbb{R}^{n_{\mathrm{tr}} \times (p+1)}$ and the test covariates matrix $X^{(\mathrm{test})} \in \mathbb{R}^{n_{\mathrm{test}} \times p}$ as context, a TFM acts as a black-box inference engine, directly outputting predictions $\widehat{\boldsymbol{y}}^{(\mathrm{test})} \in \mathbb{R}^{n_{\mathrm{test}}}$.

Despite their empirical superiority on standard benchmarks, current TFMs are inherently restricted by their single-task modeling nature. They operate under the implicit assumption that the context data originates from a single, homogeneous distribution. In practice, however, it is often necessary to integrate knowledge from multiple heterogeneous source domains to improve predictive performance on a target task—a scenario that falls squarely into the realm of transfer learning (Torrey and Shavlik, 2010). To address this limitation, we investigate the adaptation

of TFMs, specifically focusing on TabPFN, for transfer learning across heterogeneous tabular datasets.

Directly pooling source and target data into the TFM's context is usually infeasible due to two fundamental bottlenecks. First, from a statistical perspective, TFMs are highly sensitive to distribution shift; incorporating source data with a differing distribution can cause negative transfer and severely degrade the model's accuracy on the target task. Second, from an architectural perspective, models like TabPFN are strictly bounded by their maximum context size, making it impossible to ingest large-scale source datasets.

To simultaneously overcome these dual challenges of distribution shift and bounded context length, we propose a novel framework: context-constrained transfer learning via posterior-aware optimal transport distillation. Instead of indiscriminately pooling data, our method strategically distills an informative subset of source samples under size constraint to augment the target context, thereby maximizing knowledge transfer while strictly adhering to the model's architectural limits.

## 1.1  Problem Formalization

Let $\mathcal{D}^{(\mathrm{src})} = \{\boldsymbol{x}_i^{(\mathrm{src})}, y_i^{(\mathrm{src})}\}_{i=1}^{n_{\mathrm{src}}}$ be a large-scale source dataset and $\mathcal{D}^{(\mathrm{tar})} = \{\boldsymbol{x}_i^{(\mathrm{tar})}, y_i^{(\mathrm{tar})}\}_{i=1}^{n_{\mathrm{tar}}}$ be a relatively small-scale target dataset. Our goal is to make predictions for test data which are sampled from the target population. For the test data, we observe test covariates $\{\boldsymbol{x}_i^{(\mathrm{test})}\}_{i=1}^{n_{\mathrm{test}}}$ at the training phase.

**Definition 1** (TFM operator)**.** *We denote a TFM by the operator $\mathcal{P}$. Given a labeled training context $\mathcal{C}^{(\mathrm{tr})} = \{(\boldsymbol{x}_i, y_i)\}_{i=1}^{n_{\mathrm{tr}}}$ and a set of query covariates $Q = \{\boldsymbol{q}_j\}_{j=1}^{n_q}, \boldsymbol{q}_j \in \mathbb{R}^p$, the TFM outputs $\mathcal{P}(Q, \mathcal{C}^{(\mathrm{tr})}) \in \mathbb{R}^{n_q}$.*

In the above definition, $\mathcal{C}^{(\mathrm{tr})}$ provides the labeled examples used as context, whereas $Q$ contains the unlabeled covariates at which predictions are required. For instance, in the single-task target-only setting, where the source data is not available, we have query inputs $Q = X^{(\mathrm{test})}$, training context $\mathcal{C}^{(\mathrm{tr})} = \mathcal{D}^{(\mathrm{tar})}$. Thus,

$$\widehat{\boldsymbol{y}}^{(\mathrm{test})} = \mathcal{P}(X^{(\mathrm{test})}, \mathcal{D}^{(\mathrm{tar})}).$$

With source data at hand, we investigate how to summarize $\mathcal{D}^{(\mathrm{src})}$ and $\mathcal{D}^{(\mathrm{tar})}$ into a compact labeled context under the TFM context-size constraint. Due to the architectural constraints of the Transformer-based Prior-Data Fitted Network, the cardinality of the context set is strictly bounded. Specifically, for TabPFN, its constraint is $(n_{\mathrm{tr}} + n_{\mathrm{test}})p \leq 100,000$ (https://ux.priorlabs.ai/predict). In a specific task, the dimension $p$ and $n_{\mathrm{test}}$ are always pre-determined. Therefore, the constraint is essentially $n_{\mathrm{tr}} \leq n_{\max}$ for some sample size limit $n_{\max}$.

In summary, transfer learning with TFMs presents two primary challenges: the large volume and the inherent heterogeneity of source data. Because the in-context learning framework constrains the number of training samples during inference, the vast volume of source data must be efficiently summarized into a limited context window. Furthermore, the single-task nature of

current TFMs makes them sensitive to internal data heterogeneity. Consequently, it is critical to selectively incorporate only the most informative source data into the transfer learning context.

## 1.2   Our Contributions

To address the challenges of integrating large-scale heterogeneous data into TFMs, we introduce a transfer learning framework centered on posterior-aware optimal transport anchoring and local data distillation, termed TL-ANDI. Our contributions are summarized as follows.

First, we formulate source-context construction as a *budget-constrained optimal transport problem*. The proposed OT anchoring objective selects at most $n_{\max}$ source anchors by minimizing a transport cost from the observed test covariates to the selected source atoms. The cost contains two interpretable components: a covariate distance term that promotes target covariate coverage, and a posterior discrepancy term that penalizes source regions whose conditional mean is incompatible with the target task. This replaces independent heuristic sampling by a single optimization objective that is directly tied to transfer risk.

Second, after selecting the anchors, TL-ANDI locally distills source labels through kernel smoothing and then applies a residual calibration step using the target data. This preserves the black-box nature of TFMs: no task-specific model class needs to be specified, and the TFM is only queried through its standard in-context learning interface.

Third, we provide theoretical guarantees for the proposed pipeline. We establish an oracle bound for the target-context approximation error of the distilled source context, decomposing the error into transferability, source-distillation, target-pilot, and optimization components. Finally, by including the target-only predictor in the validation stage, the complete procedure enjoys a no-negative-transfer guarantee up to a validation error term.

Extensive empirical evaluations across diverse simulation setups and two real-world datasets, spanning both regression and classification tasks, demonstrate that TL-ANDI robustly mitigates negative transfer and often achieves competitive or superior performance.

## 1.3   Related Work

Transfer learning in conventional statistical models has been extensively studied recently. Recent works have studied transfer learning approaches for nonparametric regression (Cai and Wei, 2021; Reeve et al., 2021) and high-dimensional parametric models (Li et al., 2022; Tian and Feng, 2023; Li et al., 2024) among many others. However, these transfer learning methods are based on white-box models such that the functional class of the prediction model needs to be pre-specified. On the other hand, transfer learning based on black-box models, especially the TFMs in the in-context learning framework, is barely studied.

Tabular Foundation Models have emerged as a powerful alternative, utilizing in-context learning to make predictions without task-specific tuning. While early versions like TabPFN (Hollmann et al., 2025) were restricted to small-scale datasets, recent advancements have pushed these boundaries. TabPFN-2.5 (Grinsztajn et al., 2025) has scaled in-context learning capabilities to 50,000 samples and 2000 features, while Qu et al. (2025) introduce a two-stage architecture to handle up to 500,000 samples. Despite these architectural improvements, TFMs can require large computational sources when dealing with large-scale datasets and remain highly sensitive

to data heterogeneity and distribution shifts, often leading to performance degradation when source and target distributions diverge significantly. Ye et al. (2025) show that TabPFN v2's performance degrades on both large-scale and high-dimensional datasets but it can be used as a feature extractor. Ma et al. (2024) propose a method to combine in-context learning-based retrieval with self-supervised learning to train tabular foundation models.

To improve the robustness of TFMs, several adaptation strategies have been proposed. Kang et al. (2025) introduce a framework for tabular data distillation via column-embedding representations to protect privacy. Thomas et al. (2024) propose to retrieve a personalized training dataset as a local context for each test point. However, this method does not work well when posterior shift exists. Helli et al. (2024) address temporal distribution shifts by training a transformer on millions of synthetic datasets where the underlying structural causal models (SCMs) gradually change over time.

Optimal transport provides a principled language for comparing distributions and constructing representative summaries under geometric costs (Villani, 2009; Peyré and Cuturi, 2019). In contrast to classical domain-alignment methods that transport one full empirical distribution to another, our setting has a distinct context-budget constraint imposed by TFMs. This leads to a budget-constrained transport-based selection problem, where the goal is to choose a small set of source anchors that jointly cover the observed test covariates under a posterior-aware cost. The resulting objective is closely related to facility-location-type selection objectives, which motivates the greedy anchor-selection implementation used in our algorithm. Our contribution is to combine this budgeted OT view with a posterior-aware cost and source-label distillation for black-box TFM transfer learning.

## 1.4 Organization

The remainder of this paper is organized as follows. In Section 2, we introduce the methodological framework of TL-ANDI. Section 3 establishes the theoretical foundations of our approach, providing guarantees for OT anchoring, data distillation, and validation-based safe transfer. Section 4 presents extensive empirical evaluations, including regression and classification simulation studies under various distribution shift scenarios. In Section 5, we apply our proposal to two real datasets considering regression and classification tasks, respectively. We conclude with a discussion of our findings and potential avenues for future research in Section 6.

## 2 Method

We introduce the proposed method in this section. In Section 2.1, we first introduce a residual-based transfer learning framework for TFMs operating under a strict budget of $n_{\max}$ source samples. Subsequently, in Section 2.2, we detail our proposed three-stage methodology to construct a compact source context through posterior-aware OT anchoring, local label distillation, and validation-based safe aggregation.

## 2.1 Transfer Learning Using TFMs with One Informative Source

This section introduces the foundational building block of our methodology: a vanilla transfer learning approach for TFMs. In this initial formulation, we temporarily set aside the complexities associated with large-scale volume and internal source heterogeneity. We simply treat the unified source data as a single domain and model the target domain residuals to calibrate for distribution shifts. Building upon the principles established by Li et al. (2022), we propose a two-stage transfer learning procedure specifically tailored for the black-box operational nature of TFMs.

---

**Algorithm 1:** Vanilla Transfer Learning for Black-Box TFMs (Vanilla-TL)

---

**Input**: Target data $(X^{(\text{tar})}, \boldsymbol{y}^{(\text{tar})})$ of size $n_{\text{tar}}$, source data $\mathcal{D}^{(\text{src})} = (X^{(\text{src})}, \boldsymbol{y}^{(\text{src})})$ of size $n_{\text{src}}$, and test covariates $X^{(\text{test})}$.

**Output**: Predicted values $\widehat{\boldsymbol{y}}^{(\text{test})}$.

**Step 1. Learn the source prediction rule.**

Prompt the TFM, $\mathcal{P}$, using the source data as the training context, $\mathcal{C}^{(\text{tr})} = \mathcal{D}^{(\text{src})}$, and using the target and test covariates as query inputs $Q = (X^{(\text{tar})}, X^{(\text{test})})$. Denote

$$\left( \widehat{f}^{(\text{src})}(X^{(\text{tar})}), \widehat{f}^{(\text{src})}(X^{(\text{test})}) \right) = \mathcal{P}\left( (X^{(\text{tar})}, X^{(\text{test})}), \mathcal{D}^{(\text{src})} \right).$$

**Step 2. Estimate the target residual bias.**

Compute the prediction residuals on the observed target data:

$$\widehat{\boldsymbol{z}}^{(\text{tar})} = \boldsymbol{y}^{(\text{tar})} - \widehat{f}^{(\text{src})}(X^{(\text{tar})}).$$

Next, we prompt TFM using the target features and residuals as the residual-training context, $\mathcal{C}^{(\text{tr})} = (X^{(\text{tar})}, \widehat{\boldsymbol{z}}^{(\text{tar})})$, and using the test covariates as the query inputs to be predicted, $Q = X^{(\text{test})}$:

$$\widehat{\boldsymbol{u}}^{(\text{test})} = \mathcal{P}\left( X^{(\text{test})}, (X^{(\text{tar})}, \widehat{\boldsymbol{z}}^{(\text{tar})}) \right).$$

**Step 3. Final Prediction**:

$$\widehat{\boldsymbol{y}}^{(\text{test})} = \widehat{f}^{(\text{src})}(X^{(\text{test})}) + \widehat{\boldsymbol{u}}^{(\text{test})}.$$

---

Algorithm 1, which we refer to as Vanilla-TL, operates under the assumption that an informative source dataset is available and adheres to the sample size limit $n_{\text{src}} \leq n_{\max}$. In practice, however, raw source data often significantly exceeds this architectural budget and exhibits internal heterogeneity, which can precipitate negative transfer. To address these dual challenges, the following subsection details our data distillation pipeline, designed to construct a compact and highly informative source context.

## 2.2 Source Data Distillation

To construct a source context that both respects the TFM context budget and remains informative for the target task, we propose a posterior-aware optimal transport distillation procedure. The procedure contains three stages: (i) posterior-aware OT anchoring, which jointly covers the

observed test covariates and avoids source regions with large posterior mismatch; (ii) local label distillation, which replaces noisy source labels by smoothed pseudo-labels at the selected anchors; and (iii) validation-based aggregation, which selects among candidate transferred predictors and includes the target-only predictor as a safety option.

### 2.2.1 Stage 1: Posterior-Aware OT Anchor Selection

The goal of the anchoring step is to select a subset $S \subseteq [n_{\mathrm{src}}]$ with $|S| \leq n_{\max}$ from the source covariates. Instead of independently sampling source observations, we select the anchors jointly by minimizing a transport cost from the empirical test covariate distribution to the selected source atoms.

Define a locally distilled source mean at every source candidate:

$$\widetilde{y}_i^{(h)} = \frac{\sum_{r=1}^{n_{\mathrm{src}}} K_h(\boldsymbol{x}_r^{(\mathrm{src})} - \boldsymbol{x}_i^{(\mathrm{src})})y_r^{(\mathrm{src})}}{\sum_{r=1}^{n_{\mathrm{src}}} K_h(\boldsymbol{x}_r^{(\mathrm{src})} - \boldsymbol{x}_i^{(\mathrm{src})})}, \qquad i = 1, \ldots, n_{\mathrm{src}}, \tag{1}$$

where $K_h(\cdot) = h^{-p}K(\cdot/h)$ is a symmetric kernel function and $h$ denotes bandwidth. We also fit a target-only TFM predictor $\widehat{f}^{(\mathrm{tar})}$ using the calibration target data and evaluate it at the source covariates. The estimated posterior discrepancy score is then

$$\widehat{\Delta}_i^{(h)} = \widetilde{y}_i^{(h)} - \widehat{f}^{(\mathrm{tar})}(\boldsymbol{x}_i^{(\mathrm{src})}), \qquad i = 1, \ldots, n_{\mathrm{src}}. \tag{2}$$

Compared with using the raw residual $y_i^{(\mathrm{src})} - \widehat{f}^{(\mathrm{tar})}(\boldsymbol{x}_i^{(\mathrm{src})})$, the smoothed score in (2) targets the conditional-mean discrepancy and is less sensitive to source label noise.

Define the cost of transporting the $j$-th test covariate to the $i$-th source candidate as

$$\widehat{c}_{ji}^{(\lambda,h)} = \|\boldsymbol{x}_j^{(\mathrm{test})} - \boldsymbol{x}_i^{(\mathrm{src})}\|_2^2 + \lambda\{\widehat{\Delta}_i^{(h)}\}^2, \tag{3}$$

where $\lambda \geq 0$ is a tuning parameter. The first term encourages selected source anchors to cover the test covariate distribution, while the second term discourages source regions whose conditional mean is far from the target task. We select the anchor set by solving

$$\widehat{S}_{\lambda,h} \in \underset{S \subseteq [n_{\mathrm{src}}],\ |S| \leq n_{\max}}{\arg\min} \widehat{\mathcal{T}}_{\lambda,h}(S), \qquad \widehat{\mathcal{T}}_{\lambda,h}(S) = \frac{1}{n_{\mathrm{test}}} \sum_{j=1}^{n_{\mathrm{test}}} \min_{i \in S} \widehat{c}_{ji}^{(\lambda,h)}. \tag{4}$$

The objective in (4) can be viewed as a budget-constrained transport-based assignment objective from the observed test covariates to a small set of selected source anchors. Indeed, for any fixed $S$, we can show that

$$\widehat{\mathcal{T}}_{\lambda,h}(S) = \min_{\pi \geq 0} \sum_{j=1}^{n_{\mathrm{test}}} \sum_{i=1}^{n_{\mathrm{src}}} \pi_{ji}\widehat{c}_{ji}^{(\lambda,h)} \tag{5}$$

subject to $\sum_{i=1}^{n_{\mathrm{src}}} \pi_{ji} = 1/n_{\mathrm{test}}$ and $\pi_{ji} = 0$ whenever $i \notin S$. Thus, the empirical test distribution is transported to at most $n_{\max}$ selected source atoms, while avoiding source samples with large estimated posterior discrepancy.

In practice, we use a simple greedy algorithm. Let $C_0 \geq \max_{j,i} \widehat{c}_{ji}^{(\lambda,h)}$ be a finite dummy cost

and initialize $S^{(0)} = \emptyset$ and $d_j(S^{(0)}) = C_0$. At the $t$-th step, we select

$$i_t \in \underset{i \notin S^{(t-1)}}{\arg\max} \sum_{j=1}^{n_{\text{test}}} \left[ d_j(S^{(t-1)}) - \min\{d_j(S^{(t-1)}), \widehat{c}_{ji}^{(\lambda,h)}\} \right], \tag{6}$$

where $d_j(S) = \min\{C_0, \min_{r \in S} \widehat{c}_{jr}^{(\lambda,h)}\}$ and update $S^{(t)} = S^{(t-1)} \cup \{i_t\}$. This update selects the source point that yields the largest reduction in the current transport cost.

### 2.2.2 Stage 2: Local Kernel Distillation

Once the OT anchor set $\widehat{S}_{\lambda,h}$ is obtained, we form the distilled source context

$$\widetilde{\mathcal{D}}_{\lambda,h}^{(\text{src})} = \left\{ \left( \boldsymbol{x}_i^{(\text{src})}, \widetilde{y}_i^{(h)} \right) : i \in \widehat{S}_{\lambda,h} \right\}, \tag{7}$$

where $\widetilde{y}_i^{(h)}$ is defined in (1). The same local smoothing operator therefore plays two roles: it estimates posterior discrepancy for anchor selection and supplies denoised labels for the selected source anchors.

To see why this step can reduce variance, consider a generic nonparametric model $y = f(\boldsymbol{x}) + \epsilon$ with $\text{Var}(\epsilon) = \sigma_{\text{src}}^2$. The variance of $\widetilde{y}_i^{(h)}$ is approximately $\sigma_{\text{src}}^2 / m_i$, where $m_i$ is the effective number of source observations within the bandwidth $h$ around $\boldsymbol{x}_i^{(\text{src})}$. Hence, local distillation converts noisy source observations into smoother pseudo-observations while keeping the context size fixed at $n_{\max}$.

### 2.2.3 Stage 3: Aggregation for Tuning Parameter Selection

The proposed OT anchoring and distillation procedure involves two tuning parameters: the bandwidth $h \in \mathcal{H}$ and the posterior penalty $\lambda \in \Lambda$. We partition the target data $\mathcal{D}^{(\text{tar})}$ into a calibration fold $\mathcal{D}^{(\text{cal})}$ and an independent validation fold $\mathcal{D}^{(\text{val})}$. For each $(h, \lambda)$, we construct the distilled source context in (7), run Algorithm 1 using $\widetilde{\mathcal{D}}_{\lambda,h}^{(\text{src})}$ as the source data and $\mathcal{D}^{(\text{cal})}$ as the target data, and obtain a transferred predictor $\widehat{f}_{\lambda,h}$. In addition, we include the target-only predictor $\widehat{f}^{(\text{tar})}$ trained only on $\mathcal{D}^{(\text{cal})}$ as a candidate. The predictor is selected by validation error.

---

**Algorithm 2:** Transfer Learning via Posterior-Aware OT Anchoring and Distillation (TL-ANDI)

---

**Input**: Source data $\mathcal{D}^{(\mathrm{src})}$, target data $\mathcal{D}^{(\mathrm{tar})}$, test covariates $X^{(\mathrm{test})}$, context budget $n_{\max}$, bandwidth set $\mathcal{H}$, and posterior penalty set $\Lambda$.

**Output**: Predicted values $\widehat{\boldsymbol{y}}^{(\mathrm{test})}$.

**Step 0. Data splitting**:

Randomly split $\mathcal{D}^{(\mathrm{tar})}$ into two disjoint subsets: $\mathcal{D}^{(\mathrm{cal})}$ and $\mathcal{D}^{(\mathrm{val})}$, with corresponding index sets $\mathcal{N}^{(\mathrm{cal})}$ and $\mathcal{N}^{(\mathrm{val})}$. Fit a target-only predictor $\widehat{f}^{(\mathrm{tar})}$ on $\mathcal{D}^{(\mathrm{cal})}$.

**Step 1. Candidate model construction.**

**For** each $h \in \mathcal{H}$ and $\lambda \in \Lambda$ **do**:

For given $h$, compute locally distilled source labels $\widetilde{y}_i^{(h)}$ by (1). Compute posterior discrepancy scores $\widehat{\Delta}_i^{(h)}$ by (2) and costs $\widehat{c}_{ji}^{(\lambda,h)}$ by (3). Select an anchor set $\widehat{S}_{\lambda,h}$ of size at most $n_{\max}$ by the greedy update (6).

Form $\widetilde{\mathcal{D}}_{\lambda,h}^{(\mathrm{src})}$ by (7), using the anchor set $\widehat{S}_{\lambda,h}$ and the bandwidth $h$.

Execute Algorithm 1 using $\widetilde{\mathcal{D}}_{\lambda,h}^{(\mathrm{src})}$ as the source data and $\mathcal{D}^{(\mathrm{cal})}$ as the target data. Let $\widehat{f}_{\lambda,h}$ denote the resulting predictor.

**End For**

**Step 2. Validation selection.**

Select the final predictor from the candidate set

$$\mathcal{F} = \left\{ \widehat{f}^{(\mathrm{tar})} \right\} \cup \left\{ \widehat{f}_{\lambda,h} : \lambda \in \Lambda, \, h \in \mathcal{H} \right\}$$

by minimizing the validation error

$$\widehat{f}^{(\mathrm{sel})} \in \underset{\widehat{f} \in \mathcal{F}}{\arg\min} \sum_{i \in \mathcal{N}^{(\mathrm{val})}} \left( y_i^{(\mathrm{tar})} - \widehat{f}(\boldsymbol{x}_i^{(\mathrm{tar})}) \right)^2.$$

Generate the test predictions by

$$\widehat{y}_i^{(\mathrm{test})} = \widehat{f}^{(\mathrm{sel})}(\boldsymbol{x}_i^{(\mathrm{test})}), \qquad i = 1, \ldots, n_{\mathrm{test}}.$$

---

## 3 Theoretical Guarantees

In this section, we provide theoretical guarantees for the proposed procedure. We first show that the selected and distilled source context is target-compatible over the observed test covariates whenever a transferable source subset exists. Then, we prove a no-negative-transfer guarantee for the complete validation-selected procedure.

**Assumption 1.** *Assume that the source covariates $\boldsymbol{x}_i^{(\mathrm{src})} \in \mathbb{R}^p$ are drawn independently from a continuous density $p_X^{(\mathrm{src})}(\boldsymbol{x})$ supported on $[0,1]^p$, with $\inf_{\boldsymbol{x} \in [0,1]^p} p_X^{(\mathrm{src})}(\boldsymbol{x}) \geq c_0 > 0$. Assume that $y_i^{(\mathrm{src})} = f^{(\mathrm{src})}(\boldsymbol{x}_i^{(\mathrm{src})}) + \epsilon_i^{(\mathrm{src})}$, where $\epsilon_i^{(\mathrm{src})}$ are independent mean-zero sub-Gaussian errors with $\max_{i \leq n_{\mathrm{src}}} \mathbb{E}[(\epsilon_i^{(\mathrm{src})})^2] \leq \sigma_{\max}^2$.*

**Assumption 2.** *The source regression function $f^{(\mathrm{src})}$ is Hölder smooth: for some finite $\theta_1 > 0$ and $\theta_2 > 0$,*

$$|f^{(\mathrm{src})}(\boldsymbol{x}) - f^{(\mathrm{src})}(\boldsymbol{x}')| \le \theta_1 \|\boldsymbol{x} - \boldsymbol{x}'\|_2^{\theta_2}, \qquad \boldsymbol{x}, \boldsymbol{x}' \in [0,1]^p.$$

**Assumption 3.** *The kernel function $K_h(\cdot) = h^{-p} K(\cdot/h)$ satisfies that $K : \mathbb{R}^p \to \mathbb{R}_+$ is bounded, compactly supported, and*

$$\int_{\mathbb{R}^p} K(u)\, du > 0, \qquad \int_{\mathbb{R}^p} K^2(u)\, du > 0.$$

### 3.1 Quality of the Distilled Source Context

Let $f^{(\mathrm{tar})}$ denote the target regression function and define the posterior shift

$$\Delta(\boldsymbol{x}) = f^{(\mathrm{src})}(\boldsymbol{x}) - f^{(\mathrm{tar})}(\boldsymbol{x}).$$

For any source subset $S$, define the oracle posterior-aware OT cost

$$\mathcal{T}_\lambda(S) = \frac{1}{n_{\mathrm{test}}} \sum_{j=1}^{n_{\mathrm{test}}} \min_{i \in S} \left\{ \|\boldsymbol{x}_j^{(\mathrm{test})} - \boldsymbol{x}_i^{(\mathrm{src})}\|_2^2 + \lambda \Delta^2(\boldsymbol{x}_i^{(\mathrm{src})}) \right\}, \tag{8}$$

and $\mathcal{T}_{\lambda,m}^\star = \inf_{|S| \le m} \mathcal{T}_\lambda(S)$.

We measure the quality of a distilled context by the following target-context approximation error:

$$\mathfrak{D}_{\mathrm{tar}}(S, h) = \frac{1}{n_{\mathrm{test}}} \sum_{j=1}^{n_{\mathrm{test}}} \min_{i \in S} \left[ \|\boldsymbol{x}_j^{(\mathrm{test})} - \boldsymbol{x}_i^{(\mathrm{src})}\|_2^2 + \left\{ \widetilde{y}_i^{(h)} - f^{(\mathrm{tar})}(\boldsymbol{x}_j^{(\mathrm{test})}) \right\}^2 \right]. \tag{9}$$

A small value of $\mathfrak{D}_{\mathrm{tar}}(S, h)$ means that the selected source anchors, together with their distilled labels, provide a compact source context that is well aligned with the target task over the observed test covariate region.

**Lemma 3.1** (Uniform local distillation error)**.** *Suppose that Assumptions 1, 2, and 3 hold. If $h = o(1)$ and $n_{\mathrm{src}} h^p \gtrsim \log n_{\mathrm{src}}$, then with probability at least $1 - O(n_{\mathrm{src}}^{-c})$ for some constant $c > 0$,*

$$\max_{1 \le i \le n_{\mathrm{src}}} \left| \widetilde{y}_i^{(h)} - f^{(\mathrm{src})}(\boldsymbol{x}_i^{(\mathrm{src})}) \right| \le C \left\{ h^{\theta_2} + \sqrt{\frac{\log n_{\mathrm{src}}}{n_{\mathrm{src}} h^p}} \right\} =: r_{\mathrm{src}}(h),$$

*where $C > 0$ is a constant depending on the kernel, $c_0$, $\theta_1$, and $\sigma_{\max}$.*

**Assumption 4.** *The target regression function $f^{(\mathrm{tar})}$ is $L$-Lipschitz on $[0,1]^p$. Moreover, the target pilot estimator satisfies*

$$\max_{1 \le i \le n_{\mathrm{src}}} \left| \widehat{f}^{(\mathrm{tar})}(\boldsymbol{x}_i^{(\mathrm{src})}) - f^{(\mathrm{tar})}(\boldsymbol{x}_i^{(\mathrm{src})}) \right| \le r_{\mathrm{tar}}$$

*with high probability.*

For any selected set $\widehat{S}_{\lambda,h}$, define its empirical optimization residual by $\eta_m(\lambda, h) = \widehat{\mathcal{T}}_{\lambda,h}(\widehat{S}_{\lambda,h}) - \inf_{1 \le |S| \le m} \widehat{\mathcal{T}}_{\lambda,h}(S)$.

**Theorem 3.1** (Target-compatible context oracle bound)**.** *Suppose Assumptions 1–4 hold. Let* $\widehat{S}_{\lambda,h}$ *be any selected set satisfying the empirical optimization inequality*

$$\widehat{\mathcal{T}}_{\lambda,h}(\widehat{S}_{\lambda,h}) \leq \inf_{|S| \leq m} \widehat{\mathcal{T}}_{\lambda,h}(S) + \eta_m,$$

*where* $m = n_{\max}$ *and* $\eta_m \geq 0$ *is defined above. Then, with high probability,*

$$\mathfrak{D}_{\mathrm{tar}}(\widehat{S}_{\lambda,h}, h) \leq C_{\lambda,L} \left\{ \mathcal{T}^{\star}_{\lambda,m} + r^2_{\mathrm{src}}(h) + r^2_{\mathrm{tar}} + \eta_m \right\},$$

*where* $C_{\lambda,L} > 0$ *depends only on* $\lambda$ *and the Lipschitz constant* $L$*.*

Theorem 3.1 gives a direct interpretation of the proposed objective. The first term, $\mathcal{T}^{\star}_{\lambda,m}$, measures whether there exists a budget-$m$ source subset that both covers the test covariates and has small posterior discrepancy. The terms $r^2_{\mathrm{src}}(h)$ and $r^2_{\mathrm{tar}}$ quantify source-label distillation error and target-pilot estimation error. The term $\eta_m$ reflects the optimization accuracy of the numerical anchor-selection algorithm.

## 3.2 No-Negative-Transfer Guarantee

The final step of Algorithm 2 selects among transferred predictors and the target-only predictor using an independent validation fold. This yields a guarantee for the complete black-box procedure, regardless of the internal form of the TFM.

Let $\mathcal{F} = \{\widehat{f}_0, \widehat{f}_1, \ldots, \widehat{f}_M\}$ denote the finite candidate class, where $\widehat{f}_0$ is the target-only predictor $\widehat{f}^{(\mathrm{tar})}$ and the remaining candidates correspond to different values of $(\lambda, h)$. Let $R(\widehat{f}) = \mathbb{E}[(Y^{(\mathrm{tar})} - \widehat{f}(X^{(\mathrm{tar})}))^2 \mid \widehat{f}]$ be the target prediction risk.

**Theorem 3.2** (Validation oracle inequality and no negative transfer)**.** *Suppose that Assumption 1 holds and* $\max_{0 \leq k \leq M, i \in \mathcal{N}^{(\mathrm{val})}}(y_i^{(\mathrm{tar})} - \widehat{f}_k(\boldsymbol{x}_i^{(\mathrm{tar})}))^2 \leq B_n$*. For* $\widehat{f}^{(\mathrm{sel})}$ *obtained from Algorithm 2, with probability at least* $1 - \delta$*,*

$$R(\widehat{f}^{(\mathrm{sel})}) \leq \min_{0 \leq k \leq M} R(\widehat{f}_k) + 2B_n \sqrt{\frac{\log\{2(M+1)/\delta\}}{2n_{\mathrm{val}}}}.$$

*In particular, because* $\widehat{f}^{(\mathrm{tar})}$ *is the target-only predictor,*

$$R(\widehat{f}^{(\mathrm{sel})}) \leq R(\widehat{f}^{(\mathrm{tar})}) + 2B_n \sqrt{\frac{\log\{2(M+1)/\delta\}}{2n_{\mathrm{val}}}}.$$

Theorem 3.2 shows that TL-ANDI cannot be substantially worse than the target-only TFM predictor, up to the statistical uncertainty of validation selection. This provides a formal safeguard against negative transfer while retaining the ability to exploit informative source data when the OT anchoring cost is small. In practice, it is reasonable to consider $B_n = O(\log n_{\mathrm{val}})$ because $\boldsymbol{x}_i^{(\mathrm{tar})}$ and $\epsilon_i^{(\mathrm{tar})}$ are sub-Gaussian. If $\widehat{f}_k(\boldsymbol{x})$ is extremely large, it can be truncated using $\max_{i \in \mathcal{N}^{(\mathrm{val})}} |y_i^{(\mathrm{tar})}|$.

# 4 Simulation Studies

In this section, we benchmark the empirical performance of the proposed TL-ANDI framework against comparable baselines through rigorous simulation studies. Across all evaluated methods, we adopt TabPFN-2.5 as the base model. We structure our evaluation into regression simulations in Section 4.1 and classification simulations in Section 4.2. The regression simulations evaluate the overall predictive performance of TL-ANDI under homogeneous and heterogeneous source settings. Within the regression studies, the ablation experiments further validate the individual methodological stages by separately quantifying the efficacy of posterior-aware OT anchoring and the variance reduction achieved by local kernel distillation. The classification simulations examine whether TL-ANDI remains effective for binary outcomes.

For TL-ANDI across all experiments, we report the results after refitting the selected predictor. This refitting step serves as a practical implementation that further enhances the final predictive accuracy. Specifically, after the validation fold selects the final candidate in Algorithm 2, we refit the selected predictor using the full target dataset $\mathcal{D}^{(\mathrm{tar})}$ before generating the final test predictions. Appendix B.1 further examines this refitting step, and Table 6 shows that it consistently reduces the MSE across all main regression simulation settings.

Beyond the refitting analysis in Appendix B.1, we provide additional robustness and sensitivity simulation experiments in Appendix B. Specifically, Appendix B.2 compares Vanilla-TL with the Target-only baseline under different sample-size regimes. Appendix B.3 studies the sensitivity of TL-ANDI to the overlap between heterogeneous source components. Appendix B.4 investigates the effect of the target sample size on the performance of TL-ANDI.

## 4.1 Regression Simulation

To establish a rigorous comparison, we consistently evaluate three primary methods in regression experiments: the Target-only baseline, which exclusively utilizes the target data of size $n_{\mathrm{tar}}$ for prediction; TL-RAND, a naive transfer learning baseline that handles the oversized source data of size $n_{\mathrm{src}}$ by uniformly sampling a subset of budget $n_{\max}$ prior to executing Algorithm 1; and our proposed TL-ANDI method introduced in Algorithm 2. We evaluate the predictive performance by reporting the Mean Squared Error (MSE). Specifically, for test data $\{\boldsymbol{x}_i, y_i\}_{i=1}^{n_{\mathrm{test}}}$ and a generic prediction rule $\widehat{f}(\cdot)$, we report

$$\mathrm{MSE}(\widehat{f}) = \frac{1}{n_{\mathrm{test}}} \sum_{i=1}^{n_{\mathrm{test}}} (f^*(\boldsymbol{x}_i) - \widehat{f}(\boldsymbol{x}_i))^2, \tag{10}$$

where $f^*(\boldsymbol{x}_i)$ denotes the true conditional mean function for the target model.

For TL-ANDI, the posterior penalty in the OT anchoring cost is selected over $\lambda \in \{0, 0.01, 0.05, 0.1, 0.2, 0.5, 1, 2\}$, and the bandwidth $h$ is selected over the same bandwidth grid used for local kernel distillation. Specifically, we use the bandwidth grid $\mathcal{H} = \{q_{0.01}, q_{0.05}, q_{0.10}, q_{0.20}, q_{0.40}\}$, where $q_a$ denotes the $a$-quantile of pairwise Euclidean distances among standardized source covariates. Features are standardized before computing the squared Euclidean transport cost in (3), and the Epanechnikov kernel is adopted during data distillation. The target-only predictor is always included in the validation candidate set. The target dataset is partitioned into a cali-

bration subset $\mathcal{N}^{\text{(cal)}}$ containing 50% of target samples and a validation subset $\mathcal{N}^{\text{(val)}}$ containing 50% of target samples.

We consider a feature dimension of $p = 10$, a target sample size of $n_{\text{tar}} = 150$, a source sample size of $n_{\text{src}} = 20000$, a test sample size of $n_{\text{test}} = 1000$, and budget $n_{\max} \in \{500, 1000\}$.

To evaluate the proposed method under varying degrees of domain discrepancy, we design four specific experimental settings over homogeneous source data and heterogeneous source data.

### 4.1.1 Homogeneous Source

In this subsection, we consider the settings with homogeneous source data.

For the source data $\mathcal{D}^{\text{(src)}}$, the covariates are *i.i.d.* sampled from $x_{i,j}^{\text{(src)}} \sim \mathcal{U}(-1, 1)$. We consider the response model

$$y_i^{\text{(src)}} = f^{\text{(src)}}(\boldsymbol{x}_i^{\text{(src)}}) + \epsilon_i^{\text{(src)}},$$

where $\epsilon_i^{\text{(src)}} \overset{i.i.d.}{\sim} \mathcal{N}(0, 1)$ are independent noise terms. We define the underlying true function $f^{\text{(src)}}(\boldsymbol{x}_i^{\text{(src)}})$ as an additive model which contains linear, quadratic, and absolute value terms:

$$f^{\text{(src)}}(\boldsymbol{x}) = \sum_{j=1}^{10} \beta_j x_j + \sum_{j=1}^{10} \beta_{j+10} x_j^2 + \sum_{j=1}^{10} \beta_{j+20} v_j(\boldsymbol{x}) + \sum_{j=1}^{5} \beta_{j+30} |x_j|, \tag{11}$$

where $\beta_j$, $1 \leq j \leq 35$ are *i.i.d.* drawn from $\mathcal{U}(-1, 1)$ and $(v_1(\boldsymbol{x}), \ldots, v_{10}(\boldsymbol{x})) = \{x_j x_k\}_{1 \leq j < k \leq 5}$ denotes the interaction terms of the first 5 features.

For target data $\mathcal{D}^{\text{(tar)}}$, covariates $\boldsymbol{x}_i^{\text{(tar)}}$ are drawn from the same distribution as $\boldsymbol{x}_i^{\text{(src)}}$. The corresponding response $y_i^{\text{(tar)}}$ retains the standard normal additive noise structure, formulated as $y_i^{\text{(tar)}} = f^{\text{(tar)}}(\boldsymbol{x}_i^{\text{(tar)}}) + \epsilon_i^{\text{(tar)}}$, where

$$f^{\text{(tar)}}(\boldsymbol{x}) = f^{\text{(src)}}(\boldsymbol{x}) + \Delta(\boldsymbol{x}).$$

In the following, we evaluate the performance of different methods with different $\Delta(\cdot)$ functions.

(a) **Linear shift.** We first consider linear $\Delta(\boldsymbol{x})$ such that

$$\Delta(\boldsymbol{x}) = \boldsymbol{x}^T \boldsymbol{c},$$

where $c_j \sim \mathcal{U}(-1, 1)$ for $j = 1, \ldots, 5$ and $c_j = 0$ for $j \geq 6$.

(b) **Nonlinear shift.** Next, we consider a nonlinear posterior shift such that

$$\Delta(\boldsymbol{x}) = c_1 x_1 + c_2 x_2^2 + c_3 \sin(2\pi x_3) + c_4 \max(0, x_4) + c_5(x_1 x_2),$$

where the shift coefficients are sampled uniformly such that $c_j \sim \mathcal{U}(0.2, 0.8)$ for $j = 1, \ldots, 5$.

The test dataset $\mathcal{D}^{\text{(test)}}$ is generated following the identical data generating process as the target dataset $\mathcal{D}^{\text{(tar)}}$ in settings (a) and (b), respectively.

We evaluate the overall predictive performance of the proposed TL-ANDI method against the Target-only and TL-RAND baselines in settings (a) and (b) and the results are reported in

Table 1. The performance of the Target-only method remains invariant to changes in the context budget $n_{\max}$, given that it relies exclusively on the fixed target dataset $\mathcal{D}^{(\text{tar})}$. Both TL-RAND and TL-ANDI report progressively lower MSE as $n_{\max}$ increases, demonstrating their capacity to effectively leverage source data.

In Table 1, the Target-only baseline yields the highest MSE, whereas both TL-RAND and TL-ANDI successfully leverage the source data to achieve better performance. The performance gap between TL-RAND and TL-ANDI is modest under homogeneous source settings. This suggests that when the source distribution is homogeneous, even random source sampling can often transfer useful information, while TL-ANDI still provides additional gains through posterior-aware selection and label distillation. Consequently, the proposed TL-ANDI method consistently outperforms both the Target-only and TL-RAND baselines across all homogeneous source settings.

Table 1: Comparison of Mean Squared Error (MSE) among the Target-only baseline, TL-RAND, and our proposed TL-ANDI method under homogeneous source settings. The table presents the mean with standard deviation of the MSE across setting (a) linear and setting (b) nonlinear posterior shifts with context budgets of $n_{\max} \in \{500, 1000\}$. The results are summarized over 30 independent replications.

| Setting | $n_{\max}$ | Target-only | TL-RAND | TL-ANDI |
|---|---|---|---|---|
| Linear Shift | 500 | $0.5917 \pm 0.0982$ | $0.2657 \pm 0.0681$ | $\mathbf{0.2225} \pm 0.0575$ |
| | 1000 | $0.5917 \pm 0.0982$ | $0.1958 \pm 0.0504$ | $\mathbf{0.1760} \pm 0.0398$ |
| Nonlinear Shift | 500 | $0.7328 \pm 0.2098$ | $0.3533 \pm 0.0734$ | $\mathbf{0.3074} \pm 0.0649$ |
| | 1000 | $0.7328 \pm 0.2098$ | $0.2823 \pm 0.0654$ | $\mathbf{0.2685} \pm 0.0694$ |

#### 4.1.2 Heterogeneous Source

In this subsection, we consider the settings with heterogeneous source data. Specifically, for the source data $\mathcal{D}^{(\text{src})}$, we consider it to be an integration of two subpopulations.

- For $1 \leq i \leq n_{\text{src}}/2$, the source covariates are drawn from multivariate normal distributions $x_{i,j}^{(\text{src})} \overset{i.i.d.}{\sim} \mathcal{N}(0, 0.5^2)$, $1 \leq j \leq 10$. The response $y_i^{(\text{src})}$ are generated via (11) where $\beta_j$ are $i.i.d.$ generated from $\mathcal{U}(-1, 1)$. We denote the corresponding source model by $f^{(\text{src})}(\cdot)$.

- For $n_{\text{src}}/2 < i \leq n_{\text{src}}$, the source covariates are drawn from multivariate normal distributions $x_{i,j}^{(\text{src})} \overset{i.i.d.}{\sim} \mathcal{N}(1, 1)$, $1 \leq j \leq 10$. The response $y_i^{(\text{src})}$ are generated via (11) where $\beta_j$ are $i.i.d.$ generated from $\mathcal{U}(-0.5, 1.5)$. We denote the corresponding source model by $\widetilde{f}^{(\text{src})}(\cdot)$.

In the above, we consider a heterogeneous source where there exist both covariate and posterior shifts between the two components.

For the target data, we generate target domain covariates from $x_{i,j}^{(\text{tar})} \sim \mathcal{N}(0, 0.6^2)$. The target model is defined as $f^{(\text{tar})}(\boldsymbol{x}) = f^{(\text{src})}(\boldsymbol{x}) + \Delta(\boldsymbol{x})$, where $\Delta(\boldsymbol{x})$ is defined in settings (a) and (b), and $f^{(\text{src})}(\cdot)$ is the true conditional mean function for the first half of source data. We see that the target data has mild covariate shift and posterior shift from the first half of the source data but has larger distribution shift from the second half of source data. Therefore, the first half of source data are more informative than the second half and can be selected with higher

probability. The test dataset $\mathcal{D}^{(\text{test})}$ is generated following the identical data generating process as the target dataset $\mathcal{D}^{(\text{tar})}$ in settings (a) and (b), respectively.

The results are reported in Table 2. TL-RAND performs substantially worse than the Target-only baseline across both $n_{\max} = 500$ and $n_{\max} = 1000$ context budgets, demonstrating severe negative transfer. Expanding the context budget fails to meaningfully alleviate this performance degradation, highlighting the inherent risk of naive random sampling in heterogeneous source settings. In contrast, TL-ANDI consistently achieves the lowest MSE by strategically isolating informative source data from noninformative samples. Ultimately, these results highlight the advantage of TL-ANDI over both relying exclusively on limited target data and employing TL-RAND.

Table 2: Overall performance comparison of Mean Squared Error (MSE) among the Target-only baseline, TL-RAND, and our proposed TL-ANDI method under heterogeneous source settings. The table presents the mean with standard deviation of the MSE across setting (a) linear and setting (b) nonlinear posterior shifts with context budgets of $n_{\max} \in \{500, 1000\}$. The results are summarized over 30 independent replications.

| Setting | $n_{\max}$ | Target-only | TL-RAND | TL-ANDI |
|---|---|---|---|---|
| Linear Shift | 500 | $0.9780 \pm 0.2041$ | $1.6995 \pm 0.7053$ | $\mathbf{0.6471} \pm 0.2460$ |
| | 1000 | $0.9780 \pm 0.2041$ | $1.6740 \pm 0.6956$ | $\mathbf{0.5562} \pm 0.2275$ |
| Nonlinear Shift | 500 | $1.1297 \pm 0.2534$ | $1.8080 \pm 0.6959$ | $\mathbf{0.7452} \pm 0.2574$ |
| | 1000 | $1.1297 \pm 0.2534$ | $1.7499 \pm 0.6724$ | $\mathbf{0.6540} \pm 0.1891$ |

### 4.1.3 Ablation Study

In this subsection, we conduct ablation studies to evaluate the contributions of individual stages under both homogeneous and heterogeneous source settings. To explicitly evaluate the contributions of each stage, we compare the standard random baseline, TL-RAND; the random baseline augmented with our distillation process, TL-RAND w/ distillation; our proposed anchoring method without distillation, TL-ANDI w/o distillation; and our complete integrated pipeline, TL-ANDI. Results for the homogeneous and heterogeneous source settings are presented in Table 3 and Table 4, respectively.

We first examine the homogeneous source setting in Table 3. Under the homogeneous source setting, TL-RAND with distillation achieves the best performance, while TL-ANDI remains competitive compared with TL-RAND with distillation. This is because TL-RAND performs better than TL-ANDI without distillation in this setting. In this homogeneous setting, the estimated posterior-discrepancy scores do not provide strong additional information beyond random coverage, and the additional selection step may introduce mild estimation noise. As a result, TL-ANDI without distillation might perform slightly worse than TL-RAND in this homogeneous source case. The MSE reduction achieved by the complete TL-ANDI method is therefore largely attributable to the data distillation stage, as evidenced by the substantial gap in MSE between TL-ANDI without distillation and the complete TL-ANDI method.

Furthermore, we observe that the marginal improvement yielded by distillation is substantially more pronounced at $n_{\max} = 500$ than at $n_{\max} = 1000$ in Table 3. This is because the limited context leaves the model sensitive to noise under a smaller sample budget. Thus, lo-

cal kernel distillation leads to a more significant reduction in MSE. Conversely, the TabPFN model internally corrects a portion of the noise through its robust in-context learning capabilities when the budget expands to $n_{max} = 1000$. As the remaining room for improvement shrinks, the additional variance reduction provided by distillation remains beneficial but becomes less pronounced.

Table 3: Ablation study under homogeneous source conditions for setting (a) linear shift and setting (b) nonlinear shift. The table presents the mean with standard deviation of the MSE with context budgets of $n_{max} \in \{500, 1000\}$. The results are summarized over 30 independent replications.

| Shift | $n_{max}$ | Method | MSE |
|---|---|---|---|
| Linear shift | 500 | TL-RAND | $0.2657 \pm 0.0681$ |
| | | TL-RAND w/ distillation | $\mathbf{0.2148 \pm 0.0506}$ |
| | | TL-ANDI w/o distillation | $0.2813 \pm 0.0707$ |
| | | TL-ANDI | $0.2225 \pm 0.0575$ |
| | 1000 | TL-RAND | $0.1958 \pm 0.0504$ |
| | | TL-RAND w/ distillation | $\mathbf{0.1736 \pm 0.0419}$ |
| | | TL-ANDI w/o distillation | $0.2031 \pm 0.0487$ |
| | | TL-ANDI | $0.1760 \pm 0.0398$ |
| Nonlinear shift | 500 | TL-RAND | $0.3533 \pm 0.0734$ |
| | | TL-RAND w/ distillation | $\mathbf{0.3021 \pm 0.0615}$ |
| | | TL-ANDI w/o distillation | $0.3798 \pm 0.0782$ |
| | | TL-ANDI | $0.3074 \pm 0.0649$ |
| | 1000 | TL-RAND | $0.2823 \pm 0.0654$ |
| | | TL-RAND w/ distillation | $\mathbf{0.2592 \pm 0.0613}$ |
| | | TL-ANDI w/o distillation | $0.3002 \pm 0.0691$ |
| | | TL-ANDI | $0.2685 \pm 0.0694$ |

Next, we move on to the heterogeneous source setting in Table 4. We observe a distinctly different pattern that OT anchoring plays the dominant role in MSE reduction. The massive performance gap between TL-RAND and TL-ANDI without distillation in the heterogeneous source setting is significantly more pronounced than in the homogeneous source setting. This is because naive TL-RAND suffers from severe negative transfer due to its random sampling mechanism. TL-ANDI without distillation actively prevents this through posterior-aware OT anchoring, dramatically reducing the MSE. The transport cost promotes coverage of the test covariate distribution while the posterior discrepancy penalty filters out noninformative source regions. Consequently, this targeted selection avoids the substantial negative transfer that typically plagues TL-RAND. Ultimately, the complete TL-ANDI pipeline further suppresses residual label noise through its distillation mechanism, enabling the combined approach to achieve the best overall predictive performance. Overall, however, the MSE reduction achieved by OT anchoring crucially overwhelms the contribution of data distillation in the heterogeneous source setting.

Table 4: Ablation study under heterogeneous source conditions for setting (a) linear shift and setting (b) nonlinear shift. The table presents the mean with standard deviation of the MSE with context budgets of $n_{\max} \in \{500, 1000\}$. The results are summarized over 30 independent replications.

| Shift | $n_{\max}$ | Method | MSE |
|---|---|---|---|
| Linear shift | 500 | TL-RAND | $1.6995 \pm 0.7053$ |
| | | TL-RAND w/ distillation | $1.4854 \pm 0.5657$ |
| | | TL-ANDI w/o distillation | $0.6896 \pm 0.2845$ |
| | | TL-ANDI | $\mathbf{0.6471 \pm 0.2460}$ |
| | 1000 | TL-RAND | $1.6740 \pm 0.6956$ |
| | | TL-RAND w/ distillation | $1.4534 \pm 0.6004$ |
| | | TL-ANDI w/o distillation | $0.5966 \pm 0.2669$ |
| | | TL-ANDI | $\mathbf{0.5562 \pm 0.2275}$ |
| Nonlinear shift | 500 | TL-RAND | $1.8080 \pm 0.6959$ |
| | | TL-RAND w/ distillation | $1.5886 \pm 0.5653$ |
| | | TL-ANDI w/o distillation | $0.8036 \pm 0.2800$ |
| | | TL-ANDI | $\mathbf{0.7452 \pm 0.2574}$ |
| | 1000 | TL-RAND | $1.7499 \pm 0.6724$ |
| | | TL-RAND w/ distillation | $1.5707 \pm 0.6075$ |
| | | TL-ANDI w/o distillation | $0.6743 \pm 0.2123$ |
| | | TL-ANDI | $\mathbf{0.6540 \pm 0.1891}$ |

## 4.2 Classification Simulation

We further evaluate TL-ANDI in a binary classification setting. We benchmark our proposed TL-ANDI method against three baselines: the Target-only method, the TL-RAND baseline, and the TabPFN-kNN baseline proposed in Thomas et al. (2024). The TabPFN-kNN baseline operates during the prediction phase by independently retrieving the $k$-nearest source samples as a unique context for each test sample, relying strictly on covariate distances. In this simulation experiment, we fix $k = 300$. Similar to the heterogeneous regression simulation, we consider a heterogeneous source distribution consisting of two source components with identical size. We set $p = 10$, $n_{\mathrm{src}} = 20000$, $n_{\mathrm{tar}} = 150$, and $n_{\mathrm{test}} = 1000$.

Specifically, the data are generated as follows:

- For $1 \leq i \leq n_{\mathrm{src}}/2$, the source covariates are drawn from

$$x_{i,j}^{(\mathrm{src})} \overset{i.i.d.}{\sim} \mathcal{N}(0, 0.6^2), 1 \leq j \leq 10.$$

The corresponding logit model is

$$g^{(\mathrm{src})}(\boldsymbol{x}) = \sum_{j=1}^{10} \beta_j x_j + \sum_{j=1}^{10} \beta_{j+10} x_j^2 + \sum_{j=1}^{10} \beta_{j+20} v_j(\boldsymbol{x}) + \sum_{j=1}^{5} \beta_{j+30} |x_j|,$$

where $\beta_j$, $1 \leq j \leq 35$ are $i.i.d.$ drawn from $\mathcal{U}(-1, 1)$ and $(v_1(\boldsymbol{x}), \dots, v_{10}(\boldsymbol{x})) = \{x_j x_k\}_{1 \leq j < k \leq 5}$

denotes the interaction terms of the first 5 features. The binary response is generated by

$$y_i^{(\mathrm{src})} \mid \boldsymbol{x}_i^{(\mathrm{src})} \sim \mathrm{Bernoulli}\left[\sigma\left\{g^{(\mathrm{src})}(\boldsymbol{x}_i^{(\mathrm{src})})\right\}\right],$$

where $\sigma(z) = 1/(1 + \exp(-z))$ is the logistic link function.

- For $n_{\mathrm{src}}/2 < i \le n_{\mathrm{src}}$, the source covariates are also drawn from

$$x_{i,j}^{(\mathrm{src})} \overset{i.i.d.}{\sim} \mathcal{N}(1, 0.6^2), 1 \le j \le 10.$$

The corresponding logit model is $\widetilde{g}^{(\mathrm{src})}(\boldsymbol{x})$, defined as in $g^{(\mathrm{src})}(\boldsymbol{x})$ but with coefficients $\widetilde{\beta}_j$ *i.i.d.* drawn from $\mathcal{U}(-0.5, 1.5)$ .

The target and test covariates are generated from the same distribution,

$$x_{i,j}^{(\mathrm{tar})}, x_{i,j}^{(\mathrm{test})} \sim \mathcal{N}(0, 0.6^2),$$

and their responses follow the posterior model:

$$g^{(\mathrm{tar})}(\boldsymbol{x}) = 0.75 g^{(\mathrm{src})}(\boldsymbol{x}).$$

Thus,

$$y_i^{(\mathrm{tar})} \mid \boldsymbol{x}_i^{(\mathrm{tar})} \sim \mathrm{Bernoulli}\left[\sigma\left\{g^{(\mathrm{tar})}(\boldsymbol{x}_i^{(\mathrm{tar})})\right\}\right],$$
$$y_i^{(\mathrm{test})} \mid \boldsymbol{x}_i^{(\mathrm{test})} \sim \mathrm{Bernoulli}\left[\sigma\left\{g^{(\mathrm{tar})}(\boldsymbol{x}_i^{(\mathrm{test})})\right\}\right].$$

For each setting, we repeat the experiment over 30 random seeds. The evaluation metric is the mis-classification error (MCE). For the Target-only, TL-RAND, and TL-ANDI methods, the raw output provides a continuous estimate of the probability of class 1. Hyper-parameters are set identically to those used in the regression studies. We apply a standard classification threshold of 0.5 to determine the final binary prediction.

Table 5 reports the MCE over 30 random seeds. Target-only has the largest classification error and also exhibits relatively high variability, showing that the target sample size alone is not sufficient in this difficult nonlinear classification problem. The three transfer-based methods all improve over Target-only, confirming that source data are informative when incorporated properly. However, because TabPFN-kNN retrieves source samples mainly based on covariate proximity, it is not sufficient to fully resolve the posterior heterogeneity.

Overall, TL-ANDI achieves the best performance in the classification simulation experiment. More importantly, the gain over TabPFN-kNN indicates that posterior-aware source selection is beneficial in heterogeneous classification problems where misleading source samples overlap with transferable samples in covariate space.

Table 5: Mean mis-classification error of the Target-only baseline, TabPFN-kNN baseline, TL-RAND, and our proposed TL-ANDI method under the binary classification simulation experiment. The table presents the mean with standard deviation of the MCE with context budgets of $n_{\max} \in \{500, 1000\}$. The results are summarized over 30 independent replications.

| $n_{\max}$ | Target-only | TabPFN-kNN | TL-RAND | TL-ANDI |
|---|---|---|---|---|
| 500 | $0.2652 \pm 0.0431$ | $0.2411 \pm 0.0158$ | $0.1898 \pm 0.0227$ | $\mathbf{0.1520 \pm 0.0154}$ |
| 1000 | $0.2652 \pm 0.0431$ | $0.2411 \pm 0.0158$ | $0.1606 \pm 0.0173$ | $\mathbf{0.1390 \pm 0.0157}$ |

# 5 Real Data Experiments

In this section, we evaluate the performance of our proposed transfer learning algorithm in two real datasets considering prediction and classification, respectively.

## 5.1 Application to California Housing Price Dataset

The California Housing Dataset (https://www.kaggle.com/datasets/camnugent/california-housing-prices/data#) contains aggregate statistics for houses within various California districts based on 1990 census data. The dataset comprises 20,640 observations, with the median house value as the response variable and 9 features. In this analysis, we evaluate and compare the performance of three methods: the Target-only method, TL-RAND, and our proposed TL-ANDI method.

During data pre-processing, we remove all records containing missing values. We also observe that the median house value and housing median age variables are artificially capped at 500,000 and 52, respectively. We exclude these capped observations from our dataset.

To evaluate the robustness and replicability of our method, we categorize the districts based on the ocean proximity feature: "<1H OCEAN", "NEAR BAY", "INLAND", and "NEAR OCEAN". While these categories likely share structural similarities, their underlying predictive models for house values may vary, making them suitable scenarios for transfer learning. We exclude the "ISLAND" category due to its insufficient sample size, whereas the remaining four categories each contain over 2000 observations. We employ a leave-one-out strategy for our experiments. Specifically, one category is selected as the target distribution, while the data from the remaining three categories are aggregated to form the source pool in each iteration. From the source pool, we randomly sample $n_{\mathrm{src}} = 5000$ observations without replacement as the source data. From the selected target category, we randomly sample $n_{\mathrm{tar}} = 100$ observations as the target training data and another $n_{\mathrm{test}} = 1000$ observations as the test data. The context budget is fixed at $n_{\max} = 1000$. This data-splitting process is repeated 30 times to ensure statistical significance. We define the median house value as the response variable and utilize the remaining 6 features as covariates, excluding longitude and latitude to focus strictly on non-spatial attributes. All features are standardized prior to analysis, and the hyper-parameters are set identically to those used in the simulation studies.

In this experiment, performance is evaluated using the Mean Prediction Error (MPE) on the

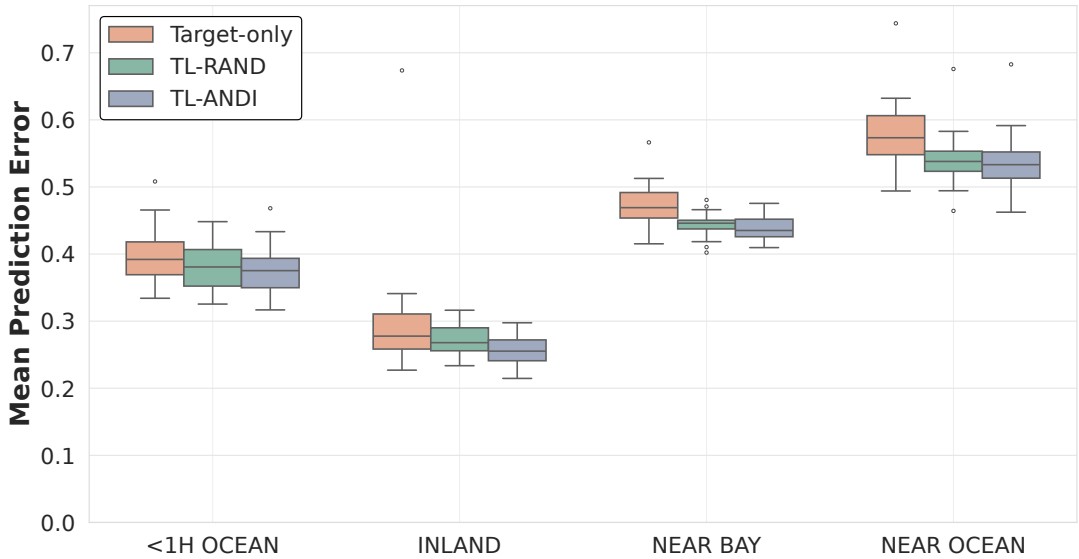

Figure 1: Mean Prediction Error (MPE) of Target-only baseline, TL-RAND, and our proposed TL-ANDI method across four target domains defined by ocean proximity. The results for each method are summarized over 30 independent replications.

test data $\{\boldsymbol{x}_i, y_i\}_{i=1}^{n_{\text{test}}}$:

$$\text{MPE}(\widehat{f}) = \frac{1}{n_{\text{test}}} \sum_{i=1}^{n_{\text{test}}} (y_i - \widehat{f}(\boldsymbol{x}_i))^2, \tag{12}$$

where $\widehat{f}(\cdot)$ denotes the prediction rule derived from each method.

Figure 1 reports the Mean Prediction Error (MPE) of the Target-only method, TL-RAND, and TL-ANDI for predicting the median house value across four target categories. Overall, TL-ANDI achieves the lowest prediction error in all four experiments, demonstrating its robustness across different target distributions. Compared with the Target-only baseline, both transfer learning methods generally benefit from incorporating source information, indicating that the source categories contain useful predictive information for the target task. However, since TL-RAND randomly selects source samples from the aggregated heterogeneous source pool, it may include samples that are less aligned with the target distribution, which weakens the transfer effect. In contrast, TL-ANDI consistently improves upon TL-RAND by selecting and distilling more informative source samples. This advantage is particularly visible for the "INLAND", "NEAR BAY", and "NEAR OCEAN" target categories, where TL-ANDI further reduces the prediction error relative to the naive random transfer baseline.

Overall, the proposed TL-ANDI method consistently achieves the lowest mean prediction errors across all ocean proximity categories. This empirical evidence validates the practical effectiveness and robustness of our proposed framework in real-world scenarios.

## 5.2 Application to Diabetes Detection

We consider a classification task in this subsection. The Diabetes Health Indicators Dataset (https://www.kaggle.com/datasets/alexteboul/diabetes-health-indicators-dataset) contains 70,692 survey responses from the CDC's 2015 Behavioral Risk Factor Surveillance System (BRFSS), a health-related telephone survey. The dataset is perfectly balanced, featuring a 50-50

split between respondents with no diabetes and those with either prediabetes or diabetes. It includes 21 feature variables detailing health-related risk behaviors, chronic health conditions, and the utilization of preventative services. In our analysis, we benchmark our proposed TL-ANDI method against three baselines: the Target-only method, the TL-RAND baseline, and TabPFN-kNN.

To evaluate our method's performance across heterogeneous data sources, we sequentially treat samples grouped by age and sex as the target domain. Given that the relationship between health risk factors and diabetes prevalence often exhibits heterogeneity across distinct demographic profiles, partitioning the dataset along the age and sex dimensions naturally induces distribution shifts, establishing a suitable transfer learning scenario. The original dataset is split into 13 groups by age. We aggregate them into three broader age clusters: young (groups 1-4), middle-aged (groups 5-8), and old (groups 9-13). By interacting these three age clusters with the binary sex variable, we construct 6 distinct demographic groups (i.e., Young female, Young male, Middle-aged female, Middle-aged male, Old female, Old male). The sample sizes across these six groups range from a minimum of 3315 to a maximum of 21418.

The response variable takes a value of 0 for individuals with no diabetes and 1 for those with prediabetes or diabetes. After defining our target and source domains based on age and sex, we select 10 key features from the remaining 19 variables to serve as covariates. For the Target-only, TL-RAND, and TL-ANDI methods, the raw output provides a continuous estimate of the probability of having diabetes. We apply a standard classification threshold of 0.5 to determine the final binary prediction. All models are evaluated and compared using mean mis-classification error.

In our experimental setup, one demographic group is isolated as the target and test domain, while all data from the other five categories are aggregated to serve as the source data. We randomly sample $n_{src} = 20000$ observations without replacement as the source data. We fix the budget $n_{max} = 1000$, the target size at $n_{tar} = 100$, and the test sample size at $n_{test} = 1000$. To ensure the robustness of our evaluation, this random sample splitting process is repeated 30 times. All continuous features are standardized prior to model training, and the hyperparameters are set identically to those utilized in the simulation studies.

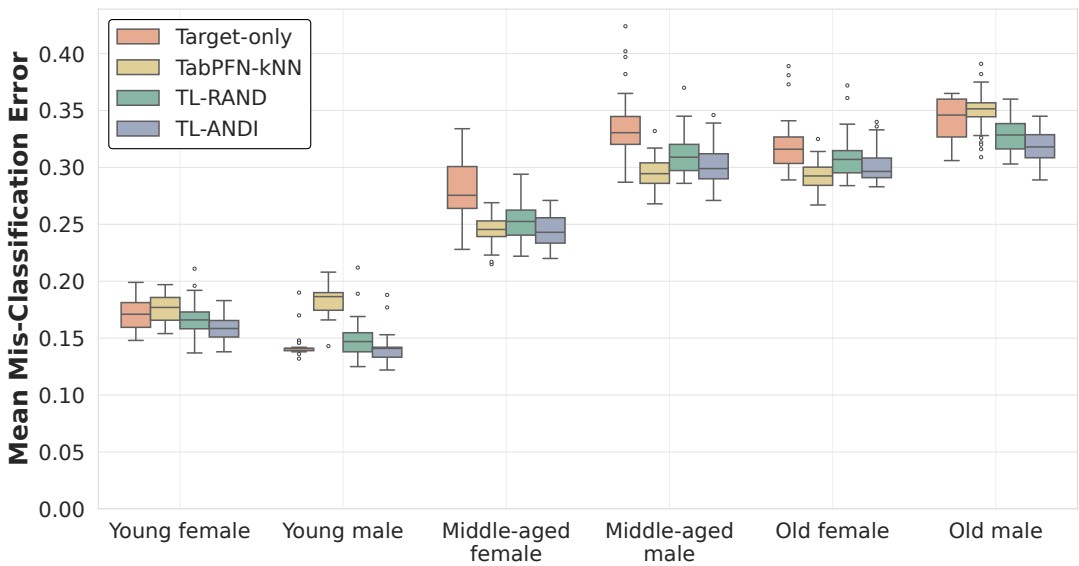

Figure 2: Mean mis-classification error of Target-only baseline, TabPFN-kNN baseline, TL-RAND, and our proposed TL-ANDI method across six demographic target domains in Diabetes Health Indicators Dataset. The results for each method are summarized over 30 independent replications.

Figure 2 reports the average mis-classification errors of the Target-only method, the TabPFN-kNN baseline, the TL-RAND baseline, and our proposed TL-ANDI method across six distinct demographic target domains. Overall, TL-ANDI demonstrates consistently stable and highly competitive or superior performance across all evaluated scenarios. Our proposal also outperforms TabPFN-kNN in many scenarios, especially in the "Young female", "Young male" and "Old male" groups.

In the "Young female" and "Young male" groups, the Target-only method demonstrates relatively small classification errors in comparison to naive transfer learning methods. TabPFN-kNN performs slightly worse in both groups, which may be due to the existence of a large distribution shift from the young-aged group to older groups so that little information can be transferred by covariate-based retrieval. Our proposal remains robust due to its careful selection of anchor points, thereby avoiding negative transfer.

In the middle-aged groups and "Old female" group, TabPFN-kNN and TL-ANDI show clear improvement. This potentially indicates that the posterior shift in these groups is mild, meaning that adjusting exclusively for the covariate shift is sufficient for successful knowledge transfer. In the "Old male" group, both TL-RAND and TL-ANDI exhibit significant performance gains through positive transfer. By carefully selecting the most informative source samples, TL-ANDI maximizes this benefit and achieves the lowest error among the four methods.

In summary, while the Target-only and TabPFN-kNN methods can achieve strong results in specific cases, their success is highly inconsistent. In comparison, TL-ANDI delivers robust and highly competitive performance across all demographic scenarios, and achieves the best or near-best error in most groups. This again demonstrates the robustness and reliability of our proposed method.

# 6 Discussion

This work introduces TL-ANDI, a framework designed to overcome the data integration bottlenecks inherent in current Tabular Foundation Models (TFMs). By formulating source-context construction as a posterior-aware budget-constrained transport-based selection problem, TL-ANDI selects a compact set of anchors that simultaneously covers the test covariate distribution and avoids source regions with large posterior mismatch. Local kernel distillation then converts these anchors into low-variance pseudo-samples, and residual calibration adapts the source prediction rule to the target task. Theoretical analysis shows that the selected and distilled source context can be target-compatible under proper conditions, while the validation step provides a no-negative-transfer safeguard by always comparing against the target-only predictor.

Despite these advantages, the computational cost of data-driven tuning and greedy OT selection remains a hurdle for real-time deployment, suggesting a need for more scalable approximations. Furthermore, as kernel-based methods are sensitive to the curse of dimensionality, future research should explore the integration of dimension reduction or sparse feature selection within the TL-ANDI framework. Developing high-dimensional transfer learning methods that maintain the efficiency of parameter-free, input-space optimization remains a vital direction for expanding the utility of TFMs in complex data environments.

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

# Appendix

## A  Proofs

*Proof of Lemma 3.1.* For any source point $\boldsymbol{x}_i^{(\text{src})}$, by (1),

$$
\begin{aligned}
\widetilde{y}_i^{(h)} - f^{(\text{src})}(\boldsymbol{x}_i^{(\text{src})}) &= \frac{\sum_{r=1}^{n_{\text{src}}} K_h(\boldsymbol{x}_r^{(\text{src})} - \boldsymbol{x}_i^{(\text{src})})\{f^{(\text{src})}(\boldsymbol{x}_r^{(\text{src})}) - f^{(\text{src})}(\boldsymbol{x}_i^{(\text{src})})\}}{\sum_{r=1}^{n_{\text{src}}} K_h(\boldsymbol{x}_r^{(\text{src})} - \boldsymbol{x}_i^{(\text{src})})} \\
&\quad + \frac{\sum_{r=1}^{n_{\text{src}}} K_h(\boldsymbol{x}_r^{(\text{src})} - \boldsymbol{x}_i^{(\text{src})})\epsilon_r^{(\text{src})}}{\sum_{r=1}^{n_{\text{src}}} K_h(\boldsymbol{x}_r^{(\text{src})} - \boldsymbol{x}_i^{(\text{src})})} \\
&:= B_i + V_i.
\end{aligned}
$$

Since $K$ is compactly supported, the numerator of $B_i$ only involves points within a distance of order $h$ from $\boldsymbol{x}_i^{(\mathrm{src})}$. By Assumption 2, uniformly over $i$,

$$|B_i| \leq C\theta_1 h^{\theta_2}.$$

By Assumptions 1 and 3, standard kernel concentration gives

$$\inf_{1 \leq i \leq n_{\mathrm{src}}} \sum_{r=1}^{n_{\mathrm{src}}} K_h(\boldsymbol{x}_r^{(\mathrm{src})} - \boldsymbol{x}_i^{(\mathrm{src})}) \geq C n_{\mathrm{src}}$$

with probability at least $1 - O(n_{\mathrm{src}}^{-c})$ when $n_{\mathrm{src}} h^p \gtrsim \log n_{\mathrm{src}}$. Conditional on the covariates, $V_i$ is sub-Gaussian with variance proxy bounded by $C/(n_{\mathrm{src}} h^p)$.

We next bound the stochastic term uniformly over all possible source anchors. Let $n = n_{\mathrm{src}}$, $X_i = \boldsymbol{x}_i^{(\mathrm{src})}$, and

$$D_i = \sum_{r=1}^{n_{\mathrm{src}}} K_h(X_r - X_i).$$

For $i = 1, \ldots, n$, define

$$w_{ir} = \frac{K_h(\boldsymbol{x}_r^{(\mathrm{src})} - \boldsymbol{x}_i^{(\mathrm{src})})}{D_i}.$$

Then

$$V_i = \sum_{r=1}^{n_{\mathrm{src}}} w_{ir} \epsilon_r^{(\mathrm{src})}.$$

We first establish a uniform lower bound for $D_i$. Under Assumptions 1 and 3, and for the usual kernels used in this paper, there exist constants $c_D > 0$ and $c > 0$ such that

$$\mathbb{P}\left(\min_{1 \leq i \leq n_{\mathrm{src}}} D_i \geq c_D n_{\mathrm{src}}\right) \geq 1 - 2 n_{\mathrm{src}} \exp(-c n_{\mathrm{src}} h^p).$$

Indeed, conditional on $\boldsymbol{x}_i^{(\mathrm{src})} = x$,

$$\mathbb{E}\left[K_h(\boldsymbol{x}_r^{(\mathrm{src})} - \boldsymbol{x})\right] = \int K_h(\boldsymbol{u} - \boldsymbol{x}) p_X^{(\mathrm{src})}(\boldsymbol{u}) \, d\boldsymbol{u} = \int K(v) p_X^{(\mathrm{src})}(\boldsymbol{x} + h\boldsymbol{v}) \, d\boldsymbol{v},$$

which is bounded below by a positive constant uniformly over $x \in [0,1]^p$, up to boundary constants, because $p_X^{(\mathrm{src})}$ is bounded away from zero on its support and $K$ has positive mass around the origin. Since $K_h(\boldsymbol{x}_r^{(\mathrm{src})} - \boldsymbol{x}) \leq \|K\|_\infty h^{-p}$, Bernstein's inequality gives

$$\mathbb{P}\left(D_i < c_D n_{\mathrm{src}} \mid \boldsymbol{x}_i^{(\mathrm{src})}\right) \leq 2 \exp(-c n_{\mathrm{src}} h^p).$$

Taking a union bound over $i = 1, \ldots, n_{\mathrm{src}}$ gives the desired uniform lower bound.

On the event

$$\mathcal{E}_X = \left\{\min_{1 \leq i \leq n} D_i \geq c_D n_{\mathrm{src}}\right\},$$

we have a uniform upper bound for the squared weights. Since $K$ is nonnegative and bounded,

$$K_h^2(\boldsymbol{u}) = h^{-2p} K^2(\boldsymbol{u}/h) \leq \|K\|_\infty h^{-p} K_h(\boldsymbol{u}).$$

Therefore, for every $i = 1, \ldots, n_{\mathrm{src}}$,

$$\sum_{r=1}^{n_{\mathrm{src}}} w_{ir}^2 = \frac{\sum_{r=1}^{n_{\mathrm{src}}} K_h^2(\boldsymbol{x}_r^{(\mathrm{src})} - \boldsymbol{x}_i^{(\mathrm{src})})}{D_i^2} \leq \frac{\|K\|_\infty h^{-p} \sum_{r=1}^{n_{\mathrm{src}}} K_h(\boldsymbol{x}_r^{(\mathrm{src})} - \boldsymbol{x}_i^{(\mathrm{src})})}{D_i^2} = \frac{\|K\|_\infty h^{-p}}{D_i} \leq \frac{C}{n_{\mathrm{src}} h^p}.$$

Conditional on $X^{(\mathrm{src})}$, $V_i$ is a weighted sum of independent sub-Gaussian errors. Hence, on $\mathcal{E}_X$, for every fixed $i$,

$$\mathbb{P}\left(|V_i| > t \mid X^{(\mathrm{src})}\right) \leq 2\exp\left(-\frac{ct^2}{\sigma_{\max}^2 \sum_{r=1}^{n_{\mathrm{src}}} w_{ir}^2}\right) \leq 2\exp\left(-\frac{cnh^p t^2}{\sigma_{\max}^2}\right).$$

Applying another union bound over $i = 1, \ldots, n_{\mathrm{src}}$, we obtain

$$\mathbb{P}\left(\max_{1 \leq i \leq n_{\mathrm{src}}} |V_i| > t \mid X^{(\mathrm{src})}\right) \leq \sum_{i=1}^n \mathbb{P}\left(|V_i| > t \mid X^{(\mathrm{src})}\right)$$
$$\leq 2n_{\mathrm{src}} \exp\left(-\frac{cn_{\mathrm{src}} h^p t^2}{\sigma_{\max}^2}\right).$$

Taking

$$t = A\sigma_{\max}\sqrt{\frac{\log n_{\mathrm{src}}}{n_{\mathrm{src}} h^p}}$$

with $A > 0$ sufficiently large gives

$$2n_{\mathrm{src}} \exp\left(-\frac{cn_{\mathrm{src}} h^p t^2}{\sigma_{\max}^2}\right) = 2n_{\mathrm{src}}^{1-cA^2} \leq 2n_{\mathrm{src}}^{-a}$$

for some constant $a > 0$. Combining this bound with the probability of $\mathcal{E}_X$, and using $n_{\mathrm{src}} h^p \gtrsim \log n_{\mathrm{src}}$, we conclude that

$$\max_{1 \leq i \leq n_{\mathrm{src}}} |V_i| \leq C\sigma_{\max}\sqrt{\frac{\log n_{\mathrm{src}}}{n_{\mathrm{src}} h^p}}$$

with probability at least $1 - Cn_{\mathrm{src}}^{-a}$. Combining the bias and stochastic terms proves the claim. $\square$

*Proof of Theorem 3.1.* Let $e_{\mathrm{src}} = r_{\mathrm{src}}(h)$ and $e_{\mathrm{tar}} = r_{\mathrm{tar}}$. On the high-probability event in Lemma 3.1 and Assumption 4,

$$|\widehat{\Delta}_i^{(h)} - \Delta(\boldsymbol{x}_i^{(\mathrm{src})})| \leq e_{\mathrm{src}} + e_{\mathrm{tar}}, \qquad i = 1, \ldots, n_{\mathrm{src}}.$$

Using $(a+b)^2 \leq 2a^2 + 2b^2$, for every $S$,

$$\mathcal{T}_\lambda(S) \leq 2\widehat{\mathcal{T}}_{\lambda,h}(S) + C_\lambda(e_{\mathrm{src}}^2 + e_{\mathrm{tar}}^2), \qquad \widehat{\mathcal{T}}_{\lambda,h}(S) \leq 2\mathcal{T}_\lambda(S) + C_\lambda(e_{\mathrm{src}}^2 + e_{\mathrm{tar}}^2).$$

As $\widehat{S}_{\lambda,h}$ satisfies the empirical optimization inequality,

$$\mathcal{T}_\lambda(\widehat{S}_{\lambda,h}) \leq 2\widehat{\mathcal{T}}_{\lambda,h}(\widehat{S}_{\lambda,h}) + C_\lambda(e_{\mathrm{src}}^2 + e_{\mathrm{tar}}^2)$$
$$\leq 2\inf_{|S| \leq m} \widehat{\mathcal{T}}_{\lambda,h}(S) + 2\eta_m + C_\lambda(e_{\mathrm{src}}^2 + e_{\mathrm{tar}}^2)$$
$$\leq 4\mathcal{T}_{\lambda,m}^\star + C_\lambda(e_{\mathrm{src}}^2 + e_{\mathrm{tar}}^2) + 2\eta_m.$$

It remains to relate $\mathfrak{D}_{\text{tar}}$ to $\mathcal{T}_\lambda$. For any pair $(j, i)$,

$$\left|\widetilde{y}_i^{(h)} - f^{(\text{tar})}(\boldsymbol{x}_j^{(\text{test})})\right| \leq \left|\widetilde{y}_i^{(h)} - f^{(\text{src})}(\boldsymbol{x}_i^{(\text{src})})\right| + |\Delta(\boldsymbol{x}_i^{(\text{src})})|$$
$$+ \left|f^{(\text{tar})}(\boldsymbol{x}_i^{(\text{src})}) - f^{(\text{tar})}(\boldsymbol{x}_j^{(\text{test})})\right|$$
$$\leq e_{\text{src}} + |\Delta(\boldsymbol{x}_i^{(\text{src})})| + L\|\boldsymbol{x}_i^{(\text{src})} - \boldsymbol{x}_j^{(\text{test})}\|_2.$$

Thus,

$$\|\boldsymbol{x}_j^{(\text{test})} - \boldsymbol{x}_i^{(\text{src})}\|_2^2 + \{\widetilde{y}_i^{(h)} - f^{(\text{tar})}(\boldsymbol{x}_j^{(\text{test})})\}^2 \leq C_{\lambda, L}\left[\|\boldsymbol{x}_j^{(\text{test})} - \boldsymbol{x}_i^{(\text{src})}\|_2^2 + \lambda\Delta^2(\boldsymbol{x}_i^{(\text{src})}) + e_{\text{src}}^2\right].$$

Taking the minimum over $i \in \widehat{S}_{\lambda, h}$ and then averaging over $j$ gives

$$\mathfrak{D}_{\text{tar}}(\widehat{S}_{\lambda, h}, h) \leq C_{\lambda, L}\{\mathcal{T}_\lambda(\widehat{S}_{\lambda, h}) + e_{\text{src}}^2\}.$$

Combining this with the previous display proves the result. $\qquad\square$

*Proof of Theorem 3.2.* Let

$$\widehat{R}_{\text{val}}(f) = \frac{1}{n_{\text{val}}} \sum_{i \in \mathcal{N}^{(\text{val})}} \left(y_i^{(\text{tar})} - f(\boldsymbol{x}_i^{(\text{tar})})\right)^2.$$

Conditional on the calibration data and source data, the candidate predictors are fixed functions of the validation covariates. By Hoeffding's inequality and a union bound over $M+1$ candidates, with probability at least $1 - \delta$,

$$\max_{0 \leq k \leq M} |\widehat{R}_{\text{val}}(\widehat{f}_k) - R(\widehat{f}_k)| \leq B_n\sqrt{\frac{\log\{2(M+1)/\delta\}}{2n_{\text{val}}}},$$

where $\widehat{R}_{\text{val}}$ denotes the empirical validation risk. Let $k^* \in \arg\min_{0 \leq k \leq M} R(\widehat{f}_k)$. Since $\widehat{f}^{(\text{sel})}$ minimizes the validation risk,

$$R(\widehat{f}^{(\text{sel})}) \leq \widehat{R}_{\text{val}}(\widehat{f}^{(\text{sel})}) + B_n\sqrt{\frac{\log\{2(M+1)/\delta\}}{2n_{\text{val}}}}$$
$$\leq \widehat{R}_{\text{val}}(\widehat{f}_{k^*}) + B_n\sqrt{\frac{\log\{2(M+1)/\delta\}}{2n_{\text{val}}}}$$
$$\leq R(\widehat{f}_{k^*}) + 2B_n\sqrt{\frac{\log\{2(M+1)/\delta\}}{2n_{\text{val}}}},$$

which proves the oracle inequality. The no-negative-transfer bound follows by choosing $k^* = 0$ or by noting that the target-only predictor is included in the candidate set. $\qquad\square$

# B  Further Simulation Results

In this section, we provide additional robustness and sensitivity simulation experiments. Specifically, Appendix B.1 further examines the refitting step after operating TL-ANDI. Appendix B.2 compares Vanilla-TL with the Target-only baseline under different sample-size regimes. Ap-

pendix B.3 studies the sensitivity of TL-ANDI to the covariate overlap between heterogeneous source components. Appendix B.4 investigates the effect of the target sample size on the performance of TL-ANDI.

## B.1 Effect of Final Refitting

In this subsection, we examine the practical effect of the final refitting step after TL-ANDI. In the proposed TL-ANDI, the target data are first split into calibration and validation folds. The calibration fold is used to construct the candidate predictors, while the validation fold is used only to select the tuning parameters and the final candidate predictor. After this validation selection step, the refitting step fixes the selected configuration and refits the corresponding prediction procedure using the full target dataset $\mathcal{D}^{(\mathrm{tar})}$ before generating test predictions. For a selected TL-ANDI predictor, we re-execute the vanilla transfer learning procedure in Algorithm 1 with the selected distilled source context and all target samples for residual calibration. For a selected target-only predictor, the target-only model is refitted using all target samples.

To evaluate the contribution of the final refitting step, we compare the Mean Squared Error (MSE) of TL-ANDI before and after final refitting under the same regression simulation settings as in Section 4.1. These settings include both homogeneous source in Section 4.1.1 and heterogeneous source in Section 4.1.2, covering linear and nonlinear distribution shifts under the main regression simulation settings. Table 6 reports the results. Across all homogeneous and heterogeneous source distribution settings, final refitting consistently reduces the MSE. This empirical improvement supports our practice of reporting the refitted TL-ANDI results in the main experiments.

Table 6: Effect of final refitting after TL-ANDI in homogeneous source and heterogeneous source with linear shift and nonlinear shift settings. The two middle columns compare the MSE before and after refitting the selected procedure using the full target dataset. The last column reports the percentage of MSE reduction due to final refitting.

| Scenario | $n_{\max}$ | MSE | | MSE Reduction (%) |
|---|---|---|---|---|
| | | Without refitting | With refitting | |
| Homogeneous linear | 500 | 0.3488 | 0.2225 | 36.21% |
| | 1000 | 0.2956 | 0.1760 | 40.46% |
| Homogeneous nonlinear | 500 | 0.3797 | 0.3074 | 19.04% |
| | 1000 | 0.3368 | 0.2685 | 20.28% |
| Heterogeneous linear | 500 | 0.7544 | 0.6471 | 14.22% |
| | 1000 | 0.6649 | 0.5562 | 16.35% |
| Heterogeneous nonlinear | 500 | 0.8222 | 0.7452 | 9.37% |
| | 1000 | 0.7222 | 0.6540 | 9.44% |

## B.2 Target-only Versus Vanilla-TL

We further conduct a sensitivity experiment to compare Target-only and Vanilla-TL, where Vanilla-TL corresponds to Algorithm 1, under homogeneous source settings. The purpose of this

experiment is to understand when the vanilla residual-based transfer procedure is beneficial.

In this experiment, we consider the homogeneous linear and homogeneous nonlinear posterior-shift settings. We vary the target sample size $n_{\mathrm{tar}} \in \{100, 150, 300, 500\}$ and the source-to-target sample size ratio $r = \frac{n_{\mathrm{src}}}{n_{\mathrm{tar}}} \in \{0.5, 1, 2, 5\}$. Thus, for each pair $(n_{\mathrm{tar}}, r)$, the source sample size is set to $n_{\mathrm{src}} = r n_{\mathrm{tar}}$. The data-generating process is the same as in the main simulations in Section 4.1.1. We compare two methods: Target-only fits the base regressor using only the target data, while Vanilla-TL first fits a source predictor using all available source samples, then fits a residual predictor on the target residuals, and finally predicts the test response by adding the estimated source prediction and target residual correction.

Figures 3 and 4 report the results for the homogeneous source with linear shift and nonlinear shift settings, respectively. Across both settings, Vanilla-TL generally improves over the Target-only baseline. This pattern is consistent with the intuition that, under homogeneous source conditions, additional source samples provide useful auxiliary information for estimating the target-domain prediction rule. Moreover, for both the Target-only predictor and Vanilla-TL, the variability of the MSE decreases as the target sample size increases.

Overall, this experiment confirms that Vanilla-TL can be useful under homogeneous source conditions, for both linear and nonlinear posterior-shift settings.

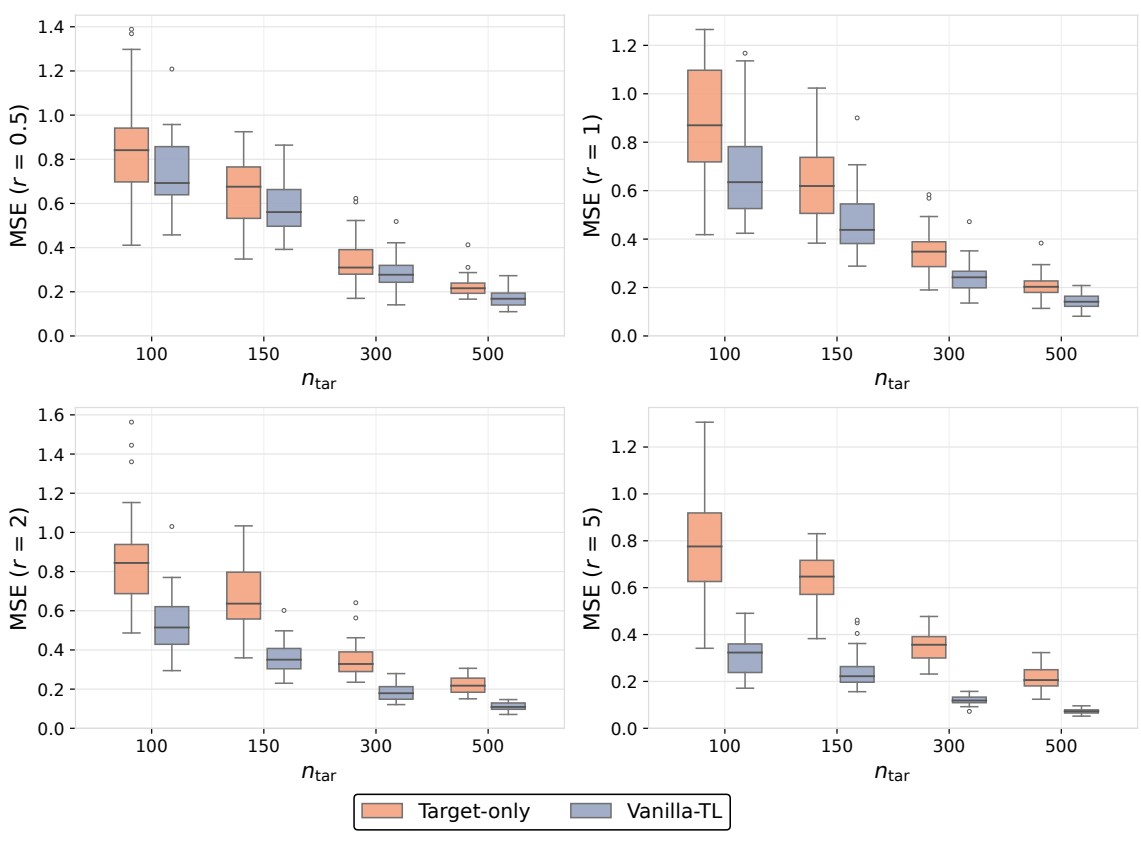

Figure 3: Mean Squared Error (MSE) of the Target-only baseline and Vanilla-TL under the homogeneous source with setting (a) linear posterior shift. The boxplots present the MSE distributions across target sample sizes $n_{\mathrm{tar}} \in \{100, 150, 300, 500\}$ and source-to-target sample ratios $r = n_{\mathrm{src}}/n_{\mathrm{tar}} \in \{0.5, 1, 2, 5\}$. The results are summarized over 30 independent replications.

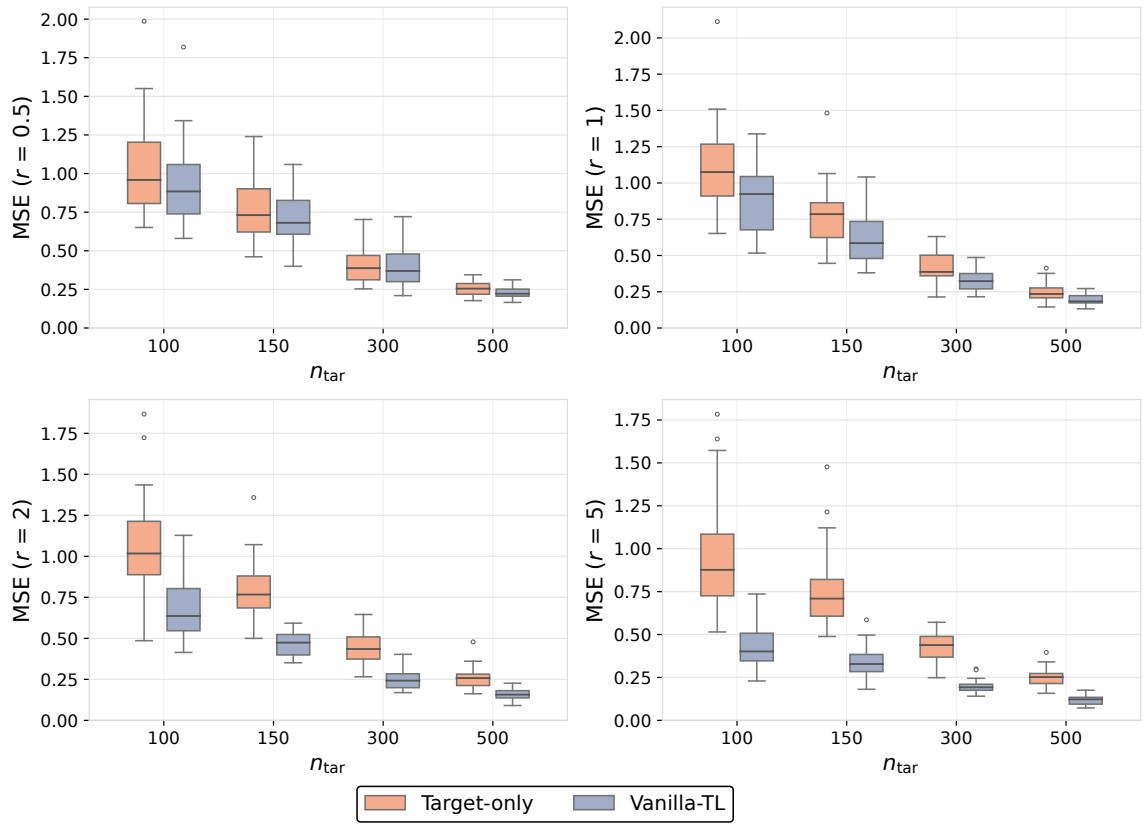

Figure 4: Mean Squared Error (MSE) of the Target-only baseline and Vanilla-TL under the homogeneous source with setting (b) nonlinear posterior shift. The boxplots present the MSE distributions across target sample sizes $n_{\text{tar}} \in \{100, 150, 300, 500\}$ and source-to-target sample size ratios $r = n_{\text{src}}/n_{\text{tar}} \in \{0.5, 1, 2, 5\}$. The results are summarized over 30 independent replications.

## B.3 Heterogeneous Source Overlap Sensitivity Analysis

To further examine the robustness of TL-ANDI under heterogeneous source distributions, we conduct an additional sensitivity experiment by varying the degree of covariate overlap between two source subpopulations.

We keep the main heterogeneous simulation design unchanged, with $p = 10$, $n_{\text{src}} = 20000$, $n_{\text{tar}} = 150$, $n_{\text{test}} = 1000$, and $n_{\max} \in \{500, 1000\}$. The source data consist of two equally sized subpopulations. For $1 \leq i \leq n_{\text{src}}/2$, the source covariates are drawn from

$$x_{i,j}^{(\text{src})} \overset{i.i.d.}{\sim} \mathcal{N}(0, 0.5^2), \qquad 1 \leq j \leq 10.$$

For $n_{\text{src}}/2 < i \leq n_{\text{src}}$, the source covariates are drawn from

$$x_{i,j}^{(\text{src})} \overset{i.i.d.}{\sim} \mathcal{N}(\mu, 1), \qquad 1 \leq j \leq 10.$$

The target and test covariates are generated in the same way as in the main heterogeneous simulation:

$$x_{i,j}^{(\text{tar})}, x_{i,j}^{(\text{test})} \overset{i.i.d.}{\sim} \mathcal{N}(0, 0.6^2), \qquad 1 \leq j \leq 10.$$

The response models for the two source subpopulations, the target and test data are also kept

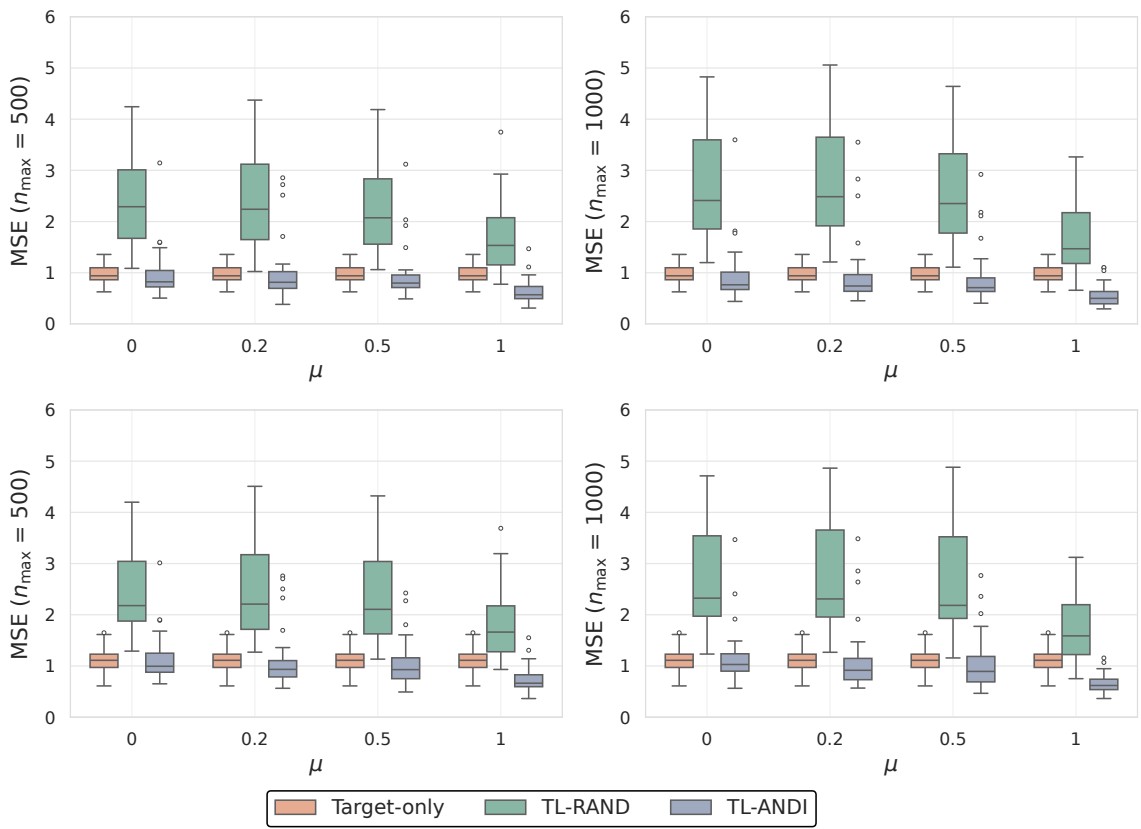

Figure 5: Mean Squared Error (MSE) among the Target-only baseline, TL-RAND, and our proposed TL-ANDI method under heterogeneous source settings with varying source-component overlap. The boxplots present the MSE distributions for setting (a) linear posterior shift (first row) and setting (b) nonlinear posterior shift (second row), across $\mu \in \{0, 0.2, 0.5, 1.0\}$. The left and right columns correspond to context budgets of $n_{\max} = 500$ and $n_{\max} = 1000$, respectively. The results are summarized over 30 independent replications.

the same as in the main heterogeneous source setting in Section 4.1.2.

We vary $\mu \in \{0, 0.2, 0.5, 1.0\}$. A smaller value of $\mu$ corresponds to stronger covariate overlap between the two source subpopulations. In particular, $\mu = 0$ represents the most challenging case, where the two source subpopulations substantially overlap in covariate space. For each setting, we repeat the experiment over 30 random seeds.

Figure 5 reports the results under the heterogeneous source with both linear and nonlinear posterior-shift settings. Across all overlap levels, TL-RAND suffers from substantially larger MSE and higher variability than the other methods, confirming that naively incorporating randomly selected source samples can lead to severe negative transfer under internal source heterogeneity. In contrast, TL-ANDI consistently achieves the lowest MSE across both linear and nonlinear settings and under both context budgets. This shows that TL-ANDI remains effective even in the high-overlap cases $\mu = 0$ and $\mu = 0.2$.

Overall, this sensitivity analysis shows that TL-ANDI is robust to source-component overlap. OT anchoring can identify source anchors that are more compatible with the target task, and the local distillation step does not break down when nearby source samples may come from different posterior distributions.

## B.4 Target Data Size Sensitivity Analysis

We further evaluate the robustness of TL-ANDI when the target sample size is limited. This experiment is motivated by the fact that the OT anchoring step relies on a target-only predictor to estimate posterior alignment. Since this predictor is trained from the available target data, it is important to examine whether TL-ANDI remains reliable when $n_{\text{tar}}$ is small.

We keep the heterogeneous simulation design in Section 4.1.2 unchanged and vary the target sample size $n_{\text{tar}} \in \{100, 150, 300\}$. The source sample size and test sample size are fixed as $n_{\text{src}} = 20000$ and $n_{\text{test}} = 1000$, respectively. We consider both heterogeneous linear and heterogeneous nonlinear posterior-shift settings, and evaluate two context budgets $n_{\text{max}} \in \{500, 1000\}$. We repeat each experiment over 30 random seeds and report the Mean Squared Error (MSE) with respect to the true conditional mean. The compared methods are Target-only, TL-RAND, and the proposed TL-ANDI. For TL-ANDI, the hyper-parameters are set identically to those utilized in the simulation studies.

Figure 6 summarizes the results. As expected, the performance of the Target-only baseline improves as $n_{\text{tar}}$ increases, since more target samples are available for fitting the target task directly. However, TL-RAND remains unstable across all target sample sizes and has the largest MSE among the three methods. In contrast, TL-ANDI is consistently robust across all values of $n_{\text{tar}}$. Even in the most challenging case $n_{\text{tar}} = 100$, where the target-only model used for posterior alignment is trained from very limited target data, TL-ANDI still achieves substantially lower MSE than TL-RAND and also improves over the Target-only baseline in both linear and nonlinear shift settings. As $n_{\text{tar}}$ increases from 100 to 300, TL-ANDI remains the best-performing method in most settings, and its variability also decreases. This suggests that the proposed OT anchoring does not overly depend on a highly accurate target-only estimator; the validation and residual calibration steps help stabilize the transfer procedure under limited target information.

Overall, the results support the reliability of TL-ANDI even with small target sample size $n_{\text{tar}}$.

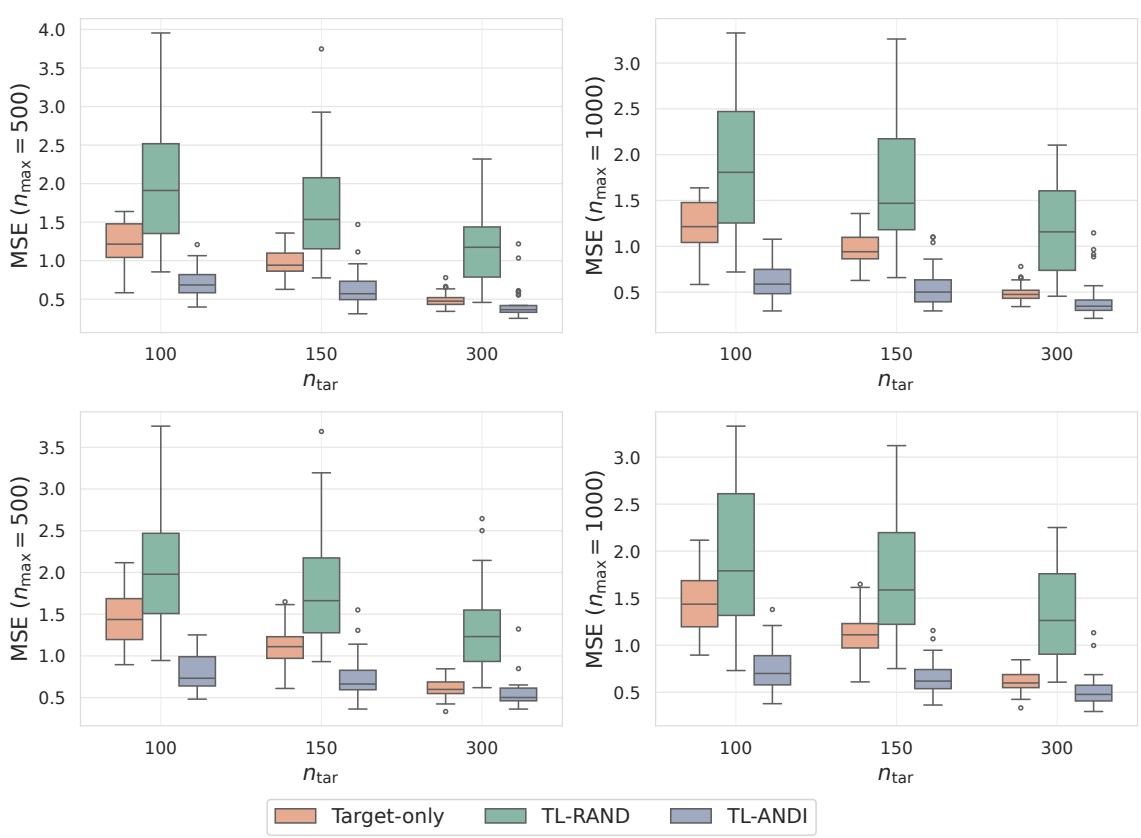

Figure 6: Mean Squared Error (MSE) of the Target-only baseline, TL-RAND, and our proposed TL-ANDI method under heterogeneous source settings with varying target sample sizes. The boxplots present the MSE distributions for setting (a) linear posterior shift (first row) and setting (b) nonlinear posterior shift (second row), across $n_{\text{tar}} \in \{100, 150, 300\}$. The left and right columns correspond to context budgets of $n_{\max} = 500$ and $n_{\max} = 1000$, respectively. The results are summarized over 30 independent replications.

