# OpenReview forum: "Context-Constrained Transfer Learning for Tabular Foundation Models via Data Distillation"
_SLADS/Section_A — Under review for SLADS_Section_A_

### Review · Reviewer_btCc · 2026-06-10

**Summary Of Contributions:**

The paper proposes TL-ANDI, a transfer learning framework for black-box Tabular Foundation Models (TFMs) such as TabPFN. Its main contribution is to adapt the context-constrained in-context tabular models to large, heterogeneous source datasets without modifying the model itself: it selects useful source anchors using covariate-shift and posterior-alignment scores, distills local source neighborhoods into smoothed pseudo-labels to reduce variance, and then applies residual calibration with target data to mitigate negative transfer. The paper also provides a theoretical bias-variance justification for the distillation step and presents simulation and real-data experiments showing that TL-ANDI improves over target-only learning and naive random source transfer, especially when source data are heterogeneous.

**Audience:**

Yes

**Claims And Evidence:**

No

**Requested Changes:**

Further discussion highlighting the challenges and the novelty of the proposed method, and some theoretical performance guarantees would be helpful.

Minor concerns:

1. Some analysis or references explaining the performance of the vanilla version, Algorithm 1, would help. For example, how does it compare with target-only learning under different target, source, and test sample sizes?

2. The operator in Equation (1) reads like a conditional probability notation, which is confusing.

**Strengths And Weaknesses:**

Strengths:

The main idea of the paper is sound and practically useful. Rather than naively pooling or randomly sampling source data, the paper proposes a coherent and useful framework that selects informative source anchors and denoises their labels before passing them to the TFM. This is particularly helpful nowadays since source data are often large, heterogeneous, and potentially harmful under a distribution shift.

Weaknesses:

1. One concern is the novelty of the proposed method. The method combines several standard ideas: importance weighting for anchor selection, local kernel smoothing for label distillation, and a residual-based transfer learning procedure. While integrating these ideas into a single pipeline is useful, the paper could better clarify what technical challenges arise in adapting them to TFMs and what is new beyond this combination. It is also not clear that the method fundamentally relies on the special structure of TFMs. The main TFM-specific motivations seem to be the context-size constraint and the black-box setting, where the model architecture or training procedure cannot be modified, so the adaptation is largely restricted to data selection and preprocessing.

2. Another concern is the design of the posterior-alignment index $\hat\alpha_i^{(PS)}$ used in the anchoring step. This step tries to select useful source points using the standard importance-weighting idea. However, this index relies on $\hat{f}^{(tar)}$, a target-only predictor trained on the small target dataset. This raises concerns about the reliability of the index, since the scarcity of target data is precisely the motivation for using transfer learning and source data in the first place.

3. The distillation step reduces variance under nice smoothness and regularity conditions. However, whether it is necessary or even useful in the heterogeneous setting is not clear. For example, in a heterogeneous source setting, nearby points in x-space may come from multiple different source distributions. If two source subpopulations overlap in covariate space but have different $\mathbb{E}[y|x]$, the local average step may seem problematic.

4. The theoretical analysis is quite limited. The authors provide only a standard proof for Lemma 3.1, which justifies the bias–variance tradeoff in the distillation step. However, there is no theoretical justification for the other components of the method, nor are there any performance guarantees for the overall algorithm.

---

> ### Author Response · Authors · 2026-07-01
> **Response to reviewer btCc (part 1)**
>
> We thank the reviewers for their careful reading and constructive comments.
> >1. One concern is the novelty of the proposed method. The method combines several standard ideas: importance weighting for anchor selection, local kernel smoothing for label distillation, and a residual-based transfer learning procedure. While integrating these ideas into a single pipeline is useful, the paper could better clarify what technical challenges arise in adapting them to TFMs and what is new beyond this combination. It is also not clear that the method fundamentally relies on the special structure of TFMs. The main TFM-specific motivations seem to be the context-size constraint and the black-box setting, where the model architecture or training procedure cannot be modified, so the adaptation is largely restricted to data selection and preprocessing.
>
> **Reply:** We thank the reviewer for raising this important concern. We have revised the manuscript to clarify that the novelty is not in kernel smoothing or residual calibration alone, but in the statistical formulation of source-context construction for context-limited black-box TFMs. The revised anchor-selection step is no longer presented as importance weighting. Instead, we formulate it as a budget-constrained transport-based selection problem with cost
> $$
> |\\boldsymbol{x}_j^{(\\textup{test})}-\\boldsymbol{x}_i^{(\\textup{src})}|_2^2
> +
> \\lambda(\\widehat{\\Delta}_i^{(h)})^2.
> $$
> Thus, the method constructs one shared, budget-feasible source context that jointly covers the observed test covariates while avoiding posterior-incompatible source regions. This differs from independent weighting or covariate-only retrieval, which do not simultaneously address context cardinality, batch-level coverage, and posterior compatibility.
>
> We also clarified that TL-ANDI is TFM-compatible rather than architecture-specific. It relies on the operational structure shared by black-box in-context tabular predictors: labeled observations constitute the inference-time context, only a finite number of observations can be supplied, test covariates are available before prediction, and the model parameters cannot be modified. Under this setting, data selection and distillation are not incidental preprocessing steps, but the main available mechanisms for post hoc transfer adaptation.
>
> The revised theory supports this formulation. Theorem 3.1 links the empirical OT anchoring objective to the quality of the target-compatible source context, while Theorem 3.2 provides a validation-based no-negative-transfer guarantee for the complete black-box procedure. We have revised the Abstract, Introduction, Method, Theory, and Discussion accordingly.
>
> >2. Another concern is the design of the posterior-alignment index $\\widehat{\\alpha}_i^{(\\textup{PS})}$ used in the anchoring step. This step tries to select useful source points using the standard importance-weighting idea. However, this index relies on $\\widehat{f}^{(\\textup{tar})}$, a target-only predictor trained on the small target dataset. This raises concerns about the reliability of the index, since the scarcity of target data is precisely the motivation for using transfer learning and source data in the first place.
>
> **Reply:** We thank the reviewer for highlighting this issue. In the revised method, the posterior signal is used only as a pilot estimate for anchor selection, not as the final transferred predictor. Its influence is stabilized in three ways: local source-label distillation reduces source noise, residual calibration corrects the transferred predictor using target data, and validation always includes the target-only predictor as a safe candidate. Therefore, if the pilot posterior signal is unreliable and leads to harmful transfer candidates, the validation step can reject them.
>
> We also added a target-sample-size sensitivity analysis in Appendix B.4. There we vary $n_{\\textup{tar}}\\in\\{100,150,300\\}$ while keeping the heterogeneous source setting fixed. TL-ANDI remains stable and consistently improves over Target-only and TL-RAND across these target sample sizes, providing empirical evidence that the proposed anchoring step remains useful even when the target pilot is trained from limited target data.

---

> ### Author Response · Authors · 2026-07-01
> **Response to reviewer btCc (part 2)**
>
> >3. The distillation step reduces variance under nice smoothness and regularity conditions. However, whether it is necessary or even useful in the heterogeneous setting is not clear. For example, in a heterogeneous source setting, nearby points in $\\boldsymbol{x}$-space may come from multiple different source distributions. If two source subpopulations overlap in covariate space but have different $\\mathbb{E}[y\\mid \\boldsymbol{x}]$, the local average step may seem problematic.
>
> **Reply:** We thank the reviewer for this helpful comment. We agree that local distillation should not be interpreted as universally safe if applied blindly to all heterogeneous source samples. TL-ANDI mitigates this issue by applying posterior-aware OT anchoring before distillation. The anchor-selection step first screens source locations using both test-covariate coverage and posterior compatibility, and the bandwidth $h$ is selected by target validation. In addition, the target-only predictor is included as a validation candidate, so harmful transferred predictors can be rejected.
>
> To examine this concern empirically, we added a heterogeneous-source overlap sensitivity analysis in Appendix B.3. The experiment varies the covariate overlap between two source subpopulations, including highly overlapping cases where nearby source samples can come from different conditional mean functions. TL-RAND becomes unstable in these settings, while TL-ANDI remains robust across overlap levels. This supports the role of posterior-aware anchoring and validation in controlling the potential risk of local averaging under source heterogeneity.
>
> >4. The theoretical analysis is quite limited. The authors provide only a standard proof for Lemma 3.1, which justifies the bias-variance tradeoff in the distillation step. However, there is no theoretical justification for the other components of the method, nor are there any performance guarantees for the overall algorithm.
>
> **Reply:** We thank the reviewer for this important comment. As also noted in our response to R1.1, the revised Section 3 now goes beyond the kernel-smoothing lemma. Theorem 3.1 analyzes the selected and distilled source context produced by the posterior-aware OT anchoring rule, and decomposes the target-context approximation error into transferability, source-distillation, target-pilot, and optimization components.
>
> We further added Theorem 3.2, a validation oracle inequality for the complete black-box procedure. Since the candidate class includes the target-only predictor, this theorem gives a no-negative-transfer guarantee up to the validation uncertainty term. These additions provide theoretical justification for the anchor-selection rule and for the overall validation-selected TL-ANDI procedure.
>
> >5. Some analysis or references explaining the performance of the vanilla version, Algorithm 1, would help. For example, how does it compare with target-only learning under different target, source, and test sample sizes?
>
> **Reply:** We thank the reviewer for this suggestion. We have clarified that Algorithm 1, now referred to as Vanilla-TL, assumes that a single informative source context is already available and satisfies the context-size constraint. TL-RAND can be viewed as its natural context-constrained implementation when the raw source dataset is oversized.
>
> We also added Appendix B.2 to compare Vanilla-TL with Target-only learning under homogeneous source settings. We vary $n_{\\textup{tar}}\\in\\{100,150,300,500\\}$ and the source-to-target ratio $r=n_{\\textup{src}}/n_{\\textup{tar}}\\in\\{0.5,1,2,5\\}$, with $n_{\\textup{test}}=1000$. The results show that Vanilla-TL generally improves over Target-only when the source data are homogeneous and informative. This also clarifies why TL-ANDI is needed: when the source pool is oversized and heterogeneous, the key challenge becomes constructing a compact transferable source context before applying residual-based transfer.
>
> >6. The operator in Equation (1) reads like a conditional probability notation, which is confusing.
>
> **Reply:** We have revised the notation in Section 1.1. We now denote a TFM by the operator $\mathcal P$. Given a labeled training context
> $
> \mathcal C^{(\textup{tr})}
> =\{(\\boldsymbol{x}_i,y_i)\}\_{i=1}^{n\_\\textup{tr}},
> $
> and a set of unlabeled prediction covariates (queries) $Q=\\{\\boldsymbol q\_j\\}\_{j=1}\^{n\_q}$, the TFM outputs $\mathcal P(Q,\mathcal C^{(\textup{tr})})\in \mathbb R^{n_q}$.
> For example, in the target-only setting,
> $$\widehat{\\boldsymbol y}^{(\\textup test)}=\mathcal P(X^{(\\textup test)},\mathcal D^{(\\textup tar)}).$$
> This avoids the previous conditional-probability-like notation and makes clear that $\mathcal P$ is a black-box prediction operator.

---

### Review · Reviewer_QH3G · 2026-06-18

**Summary Of Contributions:**

The paper studies transfer learning for tabular foundation models under context-size constraints. The proposed method, TL-ANDI, selects source anchor points using covariate and posterior alignment scores, smooths their labels through local kernel averaging, and then applies a residual correction step using target data. The paper provides a variance-reduction argument for the distillation step and reports improvements over target-only and random-transfer baselines in simulations and two real-data examples.

**Audience:**

Yes

**Broader Impact Concerns:**

The paper does not raise unusually severe ethical concerns.

**Claims And Evidence:**

Yes

**Requested Changes:**

1. Discuss the possibility of establishing theoretical guarantee for the full TL-ANDI algorithm and the challenges.
2. Clean up the potential issues of the theory pointed out above in the weakness part.
3. Add more details and discussions about Algorithm 2 as suggested above.
4. Add more benchmarks in the numerical experiment.

**Strengths And Weaknesses:**

The paper addresses a timely problem: how to use large and heterogeneous source data with tabular foundation models when the context length is limited. The proposed pipeline is intuitive, and the empirical results suggest that selective source construction can reduce negative transfer. The idea of constructing a compact source context through anchoring and local smoothing is also a natural direction for adapting in-context tabular models to transfer learning.

Weakness:
1. The theoretical guarantee is limited. Section 3 only studies the local kernel-smoothed source label in isolation. It does not give a guarantee for the full TL-ANDI predictor, the anchor sampling rule, and the residual calibration step.
2. The proof of the main theoretical result Lemma 3.1 requires some revision. The variance calculation for the kernel-smoothed noise term should account for $\sum_i K_h^2(x_i-\mathring{x}_l)/(\sum_i K_h(x_i-\mathring{x}_l))^2$. The current proof writes a bound in terms of $n_h(\mathring{x}_l)=\sum_i K_h(x_i-\mathring{x}_l)$, but this step is not justified as written. The notation also switches between $p$ and $d$. Moreover, the selected anchors should be dependent on the data which seems to be ignored in the currrent analysis.
3. Several parts of Algorithm 2 are underspecified. The bandwidth grid $\mathcal{H}$ is not given. It is also unclear whether the stochastic anchor sampling is repeated, averaged, or fixed across candidate tuning parameters. After selecting $\hat\lambda$, the algorithm appears to use only the calibration subset of the target rather than refitting with all target data, which might be not enough when $n_{\mathrm{tar}}$ is small.
4. The empirical comparisons are not yet strong enough. In simulations, the main baselines are Target-only and TL-RAND, while TabPFN-kNN is only included in the diabetes application. Retrieval-based local context methods are natural competitors for context-constrained TFMs and should be included throughout.

---

> ### Author Response · Authors · 2026-07-01
> **Response to reviewer QH3G (part 1)**
>
> We thank the reviewers for their careful reading and constructive comments.
>
> > 1. The theoretical guarantee is limited. Section 3 only studies the local kernel-smoothed source label in isolation. It does not give a guarantee for the full TL-ANDI predictor, the anchor sampling rule, and the residual calibration step.
>
> **Reply:**  We thank the reviewer for this important comment. We have substantially expanded Section 3. Lemma 3.1 now gives a uniform high-probability bound for the local source-label distillation error. Building on the revised budget-constrained optimal transport formulation, Theorem 3.1 further provides an oracle inequality for the target-context approximation error of the selected and distilled source context. This bound accounts for the best transferable source subset under the context budget, source-label distillation error, target-pilot error, and the empirical optimization residual of the anchor-selection procedure.
>
> We also added Theorem 3.2, which gives a robust guarantee for the complete black-box procedure. Since the validation candidate class contains both TL-ANDI predictors and the target-only predictor, the final selected predictor satisfies a no-negative-transfer guarantee up to the validation uncertainty term. Because the TFM is treated as a black-box prediction operator, this guarantee is formulated through validation-based safe selection rather than architecture-specific assumptions on the internal in-context learning mechanism.
>
> > 2. The proof of the main theoretical result Lemma 3.1 requires some revision. The variance calculation for the kernel-smoothed noise term should account for
> $\\frac{\\sum_i K_h^2(x_i-\\mathring{x}_l)}
> {\\left(\\sum_i K_h(x_i-\\mathring{x}_l)\\right)^2}.$
> The current proof writes a bound in terms of
> $n_h(\\mathring{x}_l)=\\sum_i K_h(x_i-\\mathring{x}_l),$
> but this step is not justified as written. The notation also switches between $p$ and $d$. Moreover, the selected anchors should be dependent on the data, which seems to be ignored in the current analysis.
>
> **Reply:** We thank the reviewer for pointing this out. We have revised the proof of Lemma 3.1 to explicitly control the variance term. With the normalized kernel $K_h(u)=h^{-p}K(u/h)$ and bounded nonnegative $K$, on the event that the kernel denominator is uniformly lower bounded, we have
> $$
> \\frac{\\sum_i K_h^2(x_i-\\mathring{x}_l)}{(\\sum_i K_h(x_i-\\mathring{x}_l))^2}
> \le
> \frac{\\|K\\|\_\infty h^{-p} \sum_i K_h(x_i-\mathring{x}_l)}{(\sum_i K_h(x_i-\mathring{x}_l))^2}
> \le
> \\frac{C} {n\_\\textup{src}h^p}.
> $$
>
> This yields the stochastic term of order $\\left(\\log n_{\\textup{src}}/(n_{\\textup{src}}h^p)\\right)^{1/2}$ after a union bound.
>
> We also removed the inconsistent notation and use $p$ throughout for the covariate dimension. Finally, to address the data dependence of the selected anchors, the revised lemma is stated uniformly over all source candidate points. Therefore, the resulting bound automatically applies to any anchor set selected from the data, including the greedy OT-selected set.
>
> > 3. Several parts of Algorithm 2 are underspecified. The bandwidth grid $\\mathcal H$ is not given. It is also unclear whether the stochastic anchor sampling is repeated, averaged, or fixed across candidate tuning parameters. After selecting $\\hat{\\lambda}$, the algorithm appears to use only the calibration subset of the target rather than refitting with all target data, which might be not enough when $n_{\\textup{tar}}$ is small.
>
> **Reply:** We thank the reviewer for pointing out these ambiguities. We have revised Algorithm 2 and the implementation description to make the tuning procedure explicit. The bandwidth grid is now specified as
> $\\mathcal H=\\{q_{0.01},q_{0.05},q_{0.10},q_{0.20},q_{0.40}\\},$
> where $q_\\alpha$ is the $\\alpha$-quantile of pairwise Euclidean distances among standardized source covariates. The posterior penalty $\\lambda$ is selected by validation over the grid reported in the experimental section.
>
> We also clarified that the revised anchor-selection step is not stochastic sampling. For each pair $(h,\\lambda)$, the anchor set is obtained by a deterministic greedy minimization of the posterior-aware transport objective, and both anchor selection and label distillation are performed inside the loop over $(h,\\lambda)$.
>
> Finally, we agree that using only the calibration fold after tuning may waste target information. In the experiments, after validation selects the final configuration, we refit the selected procedure using the full target dataset before producing the final test predictions. We added Appendix B.1 to examine this refitting step, and the results show that it consistently reduces prediction error in the main regression simulations.

---

> ### Author Response · Authors · 2026-07-01
> **Response to reviewer QH3G (part 2)**
>
> > 4. The empirical comparisons are not yet strong enough. In simulations, the main baselines are Target-only and TL-RAND, while TabPFN-kNN is only included in the diabetes application. Retrieval-based local context methods are natural competitors for context-constrained TFMs and should be included throughout.
>
> **Reply:** We thank the reviewer for this helpful suggestion. We have added a new classification simulation study in Section 4.2 that includes TabPFN-kNN as a retrieval-based local-context baseline, together with Target-only, TL-RAND, and TL-ANDI. This directly compares TL-ANDI with nearest-neighbor retrieval under the same context-constrained TFM setting.
>
> The new results show that TabPFN-kNN improves over Target-only in some cases, but remains sensitive to posterior mismatch because it retrieves source samples using covariate proximity only. TL-ANDI achieves the best performance in this simulation by combining posterior-aware OT anchoring with local label distillation. We also retain TabPFN-kNN in the diabetes real-data analysis, where TL-ANDI again achieves competitive or superior performance across demographic target groups.